# EEG-based brain-computer interface enables real-time robotic hand control at individual finger level

Yidan Ding [1], Chalisa Udompanyawit[2], Yisha Zhang[1] & Bin He [1,2,3] ✉

Brain-computer interfaces (BCIs) connect human thoughts to external devices, offering the potential to enhance life quality for individuals with motor impairments and general population. Noninvasive BCIs are accessible to a wide audience but currently face challenges, including unintuitive mappings and imprecise control. In this study, we present a real-time noninvasive robotic control system using movement execution (ME) and motor imagery (MI) of individual finger movements to drive robotic finger motions. The proposed system advances state-of-the-art electroencephalography (EEG)-BCI technology by decoding brain signals for intended finger movements into corresponding robotic motions. In a study involving 21 able-bodied experienced BCI users, we achieved real-time decoding accuracies of 80.56% for two-finger MI tasks and 60.61% for three-finger tasks. Brain signal decoding was facilitated using a deep neural network, with fine-tuning enhancing BCI performance. Our findings demonstrate the feasibility of naturalistic noninvasive robotic hand control at the individuated finger level.

Brain-computer interfaces (BCIs) have shown great promises in providing an alternative communication or control method for individuals with motor disabilities by bridging human thoughts with external devices without natural muscular outputs[1,2]. Both invasive and noninvasive BCI systems have been developed for biological limb manipulation[3–6], robotic device control[7–11], and linguistic communication[12–15]. Invasive BCI systems have achieved more sophisticated and intuitive robotic device control compared with noninvasive approaches due to higher signal-to-noise ratio and spatial resolution of recorded neural signals[16,17]. However, the requirements for surgical implantation and long-term maintenance highly limit the target populations of this approach. On the other hand, electroencephalography (EEG)-based noninvasive BCI systems have been demonstrated to achieve control of external devices in both able-bodied populations and individuals with motor impairments[18–22]. The noninvasiveness, low-cost, and portability of EEG promise to promote the practical applicability of BCI systems in both clinical and home settings.

As a promising alternative to implantable brain-controlled robotic systems, research efforts in EEG-based BCI control of upper extremity robotic devices have been ongoing for years. Meng et al.[11]

demonstrated EEG-based motor imagery (MI) BCI control of a robotic arm for reach and grasp tasks in three-dimensional space. Edelman et al.[10] demonstrated continuous control of the reach of a robotic arm using an EEG-based MI-BCI with the aid of EEG source imaging. However, the aforementioned paradigms utilize nonintuitive mappings between the user's intention and the external robotic control commands, and the control is still limited to large muscle and joint groups. In those paradigms, multi-dimensional translational motions of the robotic devices are controlled by limb motor imagery tasks, resulting in a gap between the movement intentions and the desired robotic motions. This contrasts with the natural muscle outputs through the neuromuscular pathway. Additionally, using motor intention at the limb level to actuate robotic devices also limits the precision and complexity of the systems. It therefore has driven researchers in noninvasive BCI to investigate more naturalistic control systems with a focus on more dexterous human motions.

Among various physiological functions, restoring or augmenting hand and arm function is the most desired by both motor-impaired and able-bodied populations, as even partial functional improvement

[1]Department of Biomedical Engineering, Carnegie Mellon University, Pittsburgh, PA, USA. [2]Department of Electrical and Computer Engineering, Carnegie Mellon University, Pittsburgh, PA, USA. [3]Neuroscience Institute, Carnegie Mellon University, Pittsburgh, PA, USA. ✉e-mail: bhe1@andrew.cmu.edu

can significantly extend their physical abilities[23,24]. The dexterity of the human hand allows individuals to perform precise, diverse, and flexible actions in a wide range of everyday tasks[25,26]. Its disproportionately large representation in the sensorimotor cortex underscores the hand's versatility, motivating research into decoding individual finger movements and controlling robotic devices at the finger level[27]. Impairments in upper extremity function, particularly affecting the hand, are among the most common consequences of stroke, with deficits observed in nearly half of all stroke patients[28]. These impairments significantly hinder daily living activities that demand precise, individual finger control, which conventional rehabilitation tools and gross motor BCIs often fail to address. This gap underscores the urgent need for BCI-driven robotic devices capable of decoding and restoring finger-level dexterity, thereby enabling patients to regain functionally critical fine motor skills. However, the substantial overlap in neural responses associated with individual fingers presents a key challenge for implementing precise finger-level control in such systems[29]. Efforts have been made for years to enable dexterous finger control through the use of intracortical brain signals. Hotson et al.[30] reported an electrocorticography (ECoG)-based individual finger movement real-time decoding system. Nason et al.[31] reported that the independent movements of two finger groups, the index finger and the middle-ring-small fingers combined, can be reproduced in real-time using intracortical neural signals in non-human primates. Willsey et al.[32] introduced a shallow feed-forward neural network to further improve the real-time decoding performance of the two-degree-of-freedom finger movements. Guan et al.[33] reported neural control of individual prosthetic fingers from both contralateral and ipsilateral sides in two tetraplegic patients with an implanted 96-channel array in the left posterior parietal cortex. Intracortical BCIs for quadcopter control and typing have been recently proposed using finger decoding[34,35]. Additionally, there have been attempts to noninvasively separate individual finger movements and movement intentions. Alazrai et al.[36] reported the feasibility of offline decoding of individual finger movement tasks from scalp EEG signals. The same group further explored the offline classification of finger-related motor imagery tasks from EEG signals of both able-bodied and transradial amputated subjects[37]. Lee et al.[38] used ultra-high-density EEG to perform offline decoding of the movement of individual finger pairs. Sun et al.[39] decoded non-repetitive finger flexion and extension from offline low-frequency time-domain amplitude. Alsuradi et al.[40] assessed the offline classification of individual-finger motor imagery using a data-driven approach with Shapley-informed augmentation. However, these studies relied on lengthy EEG segments and computationally intensive feature extraction processes for accurate decoding, which hindered online decoding and real-time control. Despite the promising progress in controlling individual prosthetic fingers through implantable devices, real-time control of robotic hands at the individual finger level using noninvasive neural signals has not yet been demonstrated. To bridge this critical gap, our study demonstrates the capability of real-time robotic hand control at the individual finger level utilizing a deep-learning-based EEG-BCI decoder.

Decoding individual finger movements noninvasively within the same hand presents significant challenges. Firstly, finger movements within the same hand activate relatively small and highly overlapping regions within the sensorimotor cortex, complicating the differentiation between them from noninvasive recordings[29]. Secondly, as EEG signals travel from their cerebral origin to the scalp surface, their spatial resolution and signal-to-noise ratio are significantly attenuated due to volume conduction effects and other factors, further impeding the precise decoding of neural activities[41,42]. These factors collectively pose a substantial bottleneck for effective noninvasive decoding of finger movements. However, the emergence of deep learning applications in BCI has boosted decoding performance by automatically learning hierarchical and dynamic representations from raw signals,

which holds promise for recognizing nuances in noninvasive brain signals induced by finger movements within the same hand. The benefits of deep learning techniques over conventional methods have been investigated in BCI decoding[43–45]. Lawhern et al.[46] designed EEGNet, a convolutional neural network optimized for EEG-based BCI systems. EEGNet and its variants show high versatility in various EEG-BCI applications[46,47]. By leveraging EEGNet with a fine-tuning mechanism, we were able to continuously decode single-finger movement execution (ME) and imagination of the same hand from scalp EEG signals and convert the decoding outputs into online control commands for real-time robotic finger control.

Here we present a noninvasive BCI system with the capability of continuous naturalistic robotic finger control from finger ME and MI activities. In a group of 21 able-bodied human participants who completed the study, we show that after one session of training and model fine-tuning, excellent accuracy can be achieved for 2-finger and 3-finger online robotic control using the MI and ME paradigms. We test that online training can significantly enhance task performance by integrating the network's session-specific learning through fine-tuning with the subjects' adaptation to real-time feedback, and that online smoothing can further stabilize the control outputs. The proposed system advances noninvasive BCI-based robotic control at the individual finger level, highlighting their potential to be developed into practical devices for clinical applications and everyday tasks.

## Results
In this study, we investigated finger-level real-time robotic control via EEG-based BCI. With the aim of naturalistic control, the robotic finger movement was designed to be controlled by the executed or imagined movement of the corresponding finger within the dominant hand (right hand). Twenty-one able-bodied individuals with previous limb-level BCI experience participated in one offline session and two online sessions each for finger ME and MI tasks. The offline session familiarized the participants with the task and was used to train subject-specific decoding models. In the online sessions, EEGNet-8.2 was implemented to decode individual finger movements in real time[46]. Participants received two forms of feedback reflecting the continuous decoding results, including visual feedback on a screen, where the target finger changed color to indicate the correctness of the decoding (green for correct, red for incorrect), and physical feedback from a robotic hand, which moved the detected finger in real time. The feedback period began one second after the trial onset and continued until the trial ended. To alleviate the problem of inter-session variability in decoding, in each online session, a fine-tuned model was further trained from the base model using the same-day data collected in the first half session[48]. The online task performance was evaluated using majority voting accuracy, computed as the percentage of trials in which the predicted class, determined by the majority vote of classifier outputs over multiple segments of the trial, matches the true class[49,50]. Additionally, the precision and recall with respect to each class were calculated for every classifier. Precision evaluates the model's accuracy in correctly identifying instances of a specific class, whereas recall measures the classifier's effectiveness in detecting all relevant instances. Each session included 16 runs of binary classification paradigm with thumb and pinky tasks, and 16 runs of ternary classification paradigm, decoding tasks involving the thumb, index finger, and pinky. The base model was used to decode the first 8 runs of each task, while the fine-tuned model was applied to the last 8 runs.

### Finger-level robotic control via motor intention
We demonstrated the feasibility of real-time robotic control using finger-level MI (Fig. 1, Supplementary Fig. S1, Supplementary Movies S1, S2). Across 21 participants, the results of a two-way repeated measures Analysis of Variance (ANOVA) model indicated a significant improvement in MI performance across two online sessions for both binary

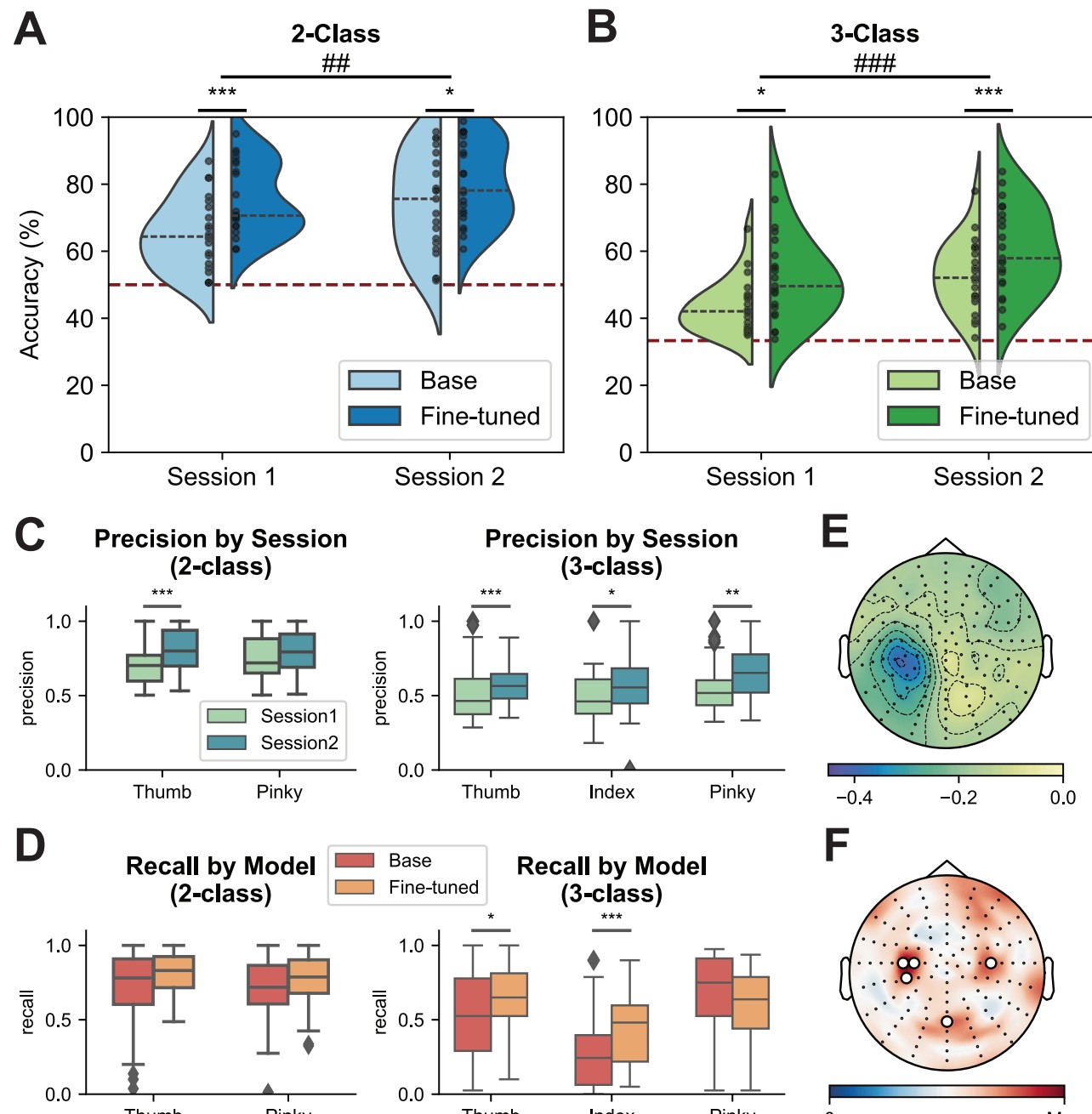

**Fig. 1 | Online performance in motor imagery (MI) robotic finger control.** Group-level online performance for 2-finger (**A**) and 3-finger (**B**) MI across sessions (*n* = 21 subjects). Comparisons between the base and fine-tuned models for Sessions 1 and 2 were conducted using a two-tailed Wilcoxon signed-rank test with Bonferroni correction for multiple comparisons (*** if *p* < 0.001, * if *p* < 0.05). Two-way repeated-measures ANOVAs were applied with main effects of session and model, and comparisons between Sessions 1 and 2 were indicated by the main effect of session (### if *p* < 0.001, ## if *p* < 0.01). The red dashed line indicates the chance level, while the gray dashed line represents the median performance under each condition. The black dots represent every instance. *P*-values: 0.0002 (**A**, Session 1), 0.0122 (**A**, Session 2), 0.0011 (**A**, Session 1 vs. 2); 0.0271 (**B**, Session 1), 3.62e−05 (**B**, Session 2), 7.55e−05 (**B**, Session 1 vs. 2). Group-level precision of 2-finger and 3-finger MI decoding across sessions (**C**) and group-level recall of

2-finger and 3-finger MI decoding results for different classifiers (**D**) (*n* = 21 subjects). The center lines indicate the median values. The boxes extend from the lower quartile to the upper quartile. Diamonds indicate outliers that are more than 1.5 times the interquartile range above the third quartile or below the first quartile. Comparisons between the precision across sessions and recall across models were conducted using a two-tailed Wilcoxon signed-rank test with Bonferroni correction for multiple comparisons (*** if *p* < 0.001, ** if *p* < 0.01, * if *p* < 0.05). *P*-values: 4.62e−08 (**C**, Thumb, 2-class), 0.0009 (**C**, Thumb, 3-class), 0.0395 (**C**, Index, 3-class), 0.0012 (**C**, Pinky, 3-class), 0.0412 (**D**, Thumb, 3-class), 0.0007 (**D**, Index, 3-class). **E** Group averaged alpha band (8–13 Hz) ERD topology for MI online control tasks. **F** Group-averaged saliency topological maps of the EEGNet models for online MI sessions. Channels with the highest 25% saliency values are highlighted by white circles on the electrodes.

(*F* = 14.455, *p* = 0.001, Fig. 1A) and ternary (*F* = 24.590, *p* < 0.001, Fig. 1B) paradigms. Performance enhancements were evident in the precision and recall metrics for individual classes, showing consistent improvements across all categories between sessions (Fig. 1C, Supplementary

Fig. S1A). Furthermore, the feature representations learned by the deep learning decoder exhibited enhanced class discriminability with increased training size across sessions, as demonstrated by visualizations of intermediate-layer activations (Supplementary Fig. S2).

The ANOVA also identified a significant main effect of model type for binary ($F = 31.380$, $p < 0.001$) and ternary ($F = 22.443$, $p < 0.001$) tasks. Post-hoc pairwise comparisons were performed to assess the efficacy of fine-tuning during each session on the performance outcomes. Bonferroni-corrected two-tailed Wilcoxon signed-rank test results indicated that fine-tuned models significantly outperformed base models in all online sessions and paradigms ($p < 0.001$ for Session 1 binary tasks and Session 2 ternary tasks, $p < 0.05$ for Session 2 binary tasks and Session 1 ternary tasks; Fig. 1A, B). Comparisons of intermediate-layer feature maps further indicated that fine-tuned models exhibited enhanced decoding capabilities relative to base models (Supplementary Fig. S2). This observation aligns with previous studies demonstrating fine-tuning effectively enhances model capacity by adapting it to the target dataset's distribution[51,52]. Despite relatively balanced precision values across all categories using the base models (Supplementary Fig. S1B), imbalances were noted in the models' ability to identify relevant instances of different classes (Supplementary Fig. S1C, D). Specifically, the thumb and the index finger in the 3-class paradigm had lower recall values when using the base models (Fig. 1D). Fine-tuning contributed to addressing these class imbalances by significantly enhancing the detection capability of the categories with low recall values, as confirmed by multiple-comparison corrected two-tailed Wilcoxon signed-rank test results (Fig. 1D). After one session of online training and model fine-tuning, 2-finger online MI robotic control can be achieved with an average accuracy of 80.56% across 21 subjects, and 3-finger online control can achieve the average accuracy of 60.61%.

The involvement of contralateral motor and premotor cortices during MI has been documented in many studies[53,54]. To further elucidate this phenomenon, alpha band (8–13 Hz) event-related desynchronization (ERD) patterns were extracted from EEG data to analyze MI-induced electrophysiological activation. A significant power decrease was observed in the left sensorimotor region during MI tasks (Fig. 1E). Complementing these findings, saliency maps generated by EEGNet models, using the gradient of the output class probability relative to the input EEG signals, highlighted the hand knob area in the left primary motor cortex (Fig. 1F). These findings align with prior research identifying this region as critical for motor intention. The spatial concordance between the saliency topologies of the decoders and the ERD patterns reinforced the sensorimotor region's role in the model's predictions. To validate this observation, we conducted offline decoding experiments comparing performance across sensorimotor-specific channels, non-sensorimotor channels, and whole-scalp inputs. To control for the potential confounding effect of input dimensionality, the number of channels in the sensorimotor and non-sensorimotor regions was matched. Sensorimotor channels achieved decoding performance comparable to whole-scalp input, whereas non-sensorimotor channels showed significantly reduced accuracy (Supplementary Fig. S3A, C), reinforcing the critical role of sensorimotor regions in finger MI decoding. Further dissection of cortical contributions was achieved by isolating four subregions, each comprising 11 channels, including contralateral and ipsilateral hand knob areas, which are responsible for controlling the fine movements of fingers, occipital cortex, and frontal regions (Supplementary Fig. S4A). Among the subregions, the left and right hand knob areas yielded the highest decoding accuracies (Supplementary Fig. S4E), aligning with our electrophysiological analysis results and confirming the decoder's dominant reliance on these regions. This convergence confirms the decoder's primary reliance on the sensorimotor region during MI. However, the accuracy using only the contralateral hand knob as input was found to be significantly lower than 128-channel whole-scalp input (Supplementary Fig. S4C). This finding aligns with the saliency maps of the EEGNet models, which revealed a predominant focus on the hand knob area in the left primary motor cortex with additional emphasis in the right hand knob region and parietal channels (Fig. 1F). Together,

the saliency maps and subregion analyses underscore the key role of the contralateral hand knob region in decoding finger movements, while also suggesting that distributed electrode configurations capture supplementary task-relevant signals.

To delineate the factors contributing to the observed performance improvements, offline simulations were conducted using EEG data collected during MI robotic control sessions. Data from fine-tuned runs in Session 2 were evaluated using three models: the Session 1 Base model (S1-B), the Session 1 Fine-tuned model (S1-F), and the Session 2 Base model (S2-B). Across sessions and models, significant or near-significant improvements were evident in binary tasks (S1-B vs. S1-F: $p = 0.052$; S2-B vs. S2-F: $p < 0.001$; S1-B vs. S2-B: $p = 0.019$; S1-F vs. S2-F: $p < 0.001$, Fig. 2A) and ternary tasks (S1-B vs. S1-F: $p = 0.044$; S2-B vs. S2-F: $p = 0.004$; S1-B vs. S2-B: $p < 0.001$; S1-F vs. S2-F: $p < 0.001$, Fig. 2A). The observed improvements were achieved using the same input data across compared models, ruling out data quality as confounding factors, and therefore highlighting the machine learning effects. The enhanced performance of fine-tuned models (S1-F, S2-F) over their base counterparts (S1-B, S2-B) demonstrates the effectiveness of machine learning-driven adaptation in refining feature extraction for session-specific data. Furthermore, the superior performance of Session 2 models (S2-B, S2-F) over Session 1 models (S1-B, S1-F) suggests enhanced decoding capacities facilitated by increased training sizes. To investigate the effects of human learning, offline decoding was performed using the Filter Bank Common Spatial Pattern (FBCSP) method paired with a Linear Discriminant Analysis (LDA) classifier[50]. A two-way repeated-measures ANOVA revealed a significant main effect of the model in 3-class offline simulations (Fig. 2B). Significant performance improvement was observed during the first 3-class session when comparing the last eight fine-tuned runs with the first eight base runs ($p < 0.01$, Fig. 2B). A numerical enhancement in decoding performance was also noted in Session 2 compared to Session 1 for ternary tasks (Cohen's $d = 0.222$, $F = 3.065$, $p = 0.095$). However, no clear improvement trend was observed in online decoding accuracy over eight continuous runs (Fig. 2C). These findings suggest an initial within-session human learning effect and cross-session human training for the 3-finger MI tasks.

## Movement execution controlled robotic finger movements

The same group of participants ($n = 21$) also participated in one offline session and two online sessions of finger ME tasks, following the same experimental design as the MI tasks (Supplementary Movies S3, S4). Significant performance improvements were observed across sessions and between the fine-tuned and base models, as indicated by a two-way repeated-measures ANOVA (2-class session effect: $F = 8.826$, $p = 0.007$; 2-class model effect: $F = 35.034$, $p < 0.001$, 3-class session effect: $F = 12.914$, $p = 0.001$; 3-class model effect: $F = 35.869$, $p < 0.001$, Fig. 3A, B, Supplementary Fig. S5). Enhanced class separability was observed in the learned feature space across sessions, with fine-tuned models exhibiting clearer class distinction than their base model counterparts (Supplementary Fig. S6). Online training substantially increased precision and recall for thumb and pinky across sessions (Fig. 3C, Supplementary Fig. S5A). Offline simulation results comparing different models on the same dataset demonstrated enhanced decoding performance using Session 2 models compared to Session 1 models (2-class decoding: p(S1-B vs. S2-B) = 0.140; p(S1-F vs. S2-F) < 0.001; 3-class decoding: p(S1-B vs. S2-B) = 0.009; p(S1-F vs. S2-F) < 0.001, Supplementary Fig. S7A). This improvement highlighted the impact of increased training size on machine learning performance across sessions. Significant decoding improvements were observed when comparing fine-tuned models to base models within Session 2 (2-class decoding: p(S2-B vs. S2-F) < 0.001; 3-class decoding: p(S2-B vs. S2-F) = 0.001, Supplementary Fig. S7A), confirming the efficacy of fine-tuning in enhancing performance within the same session.

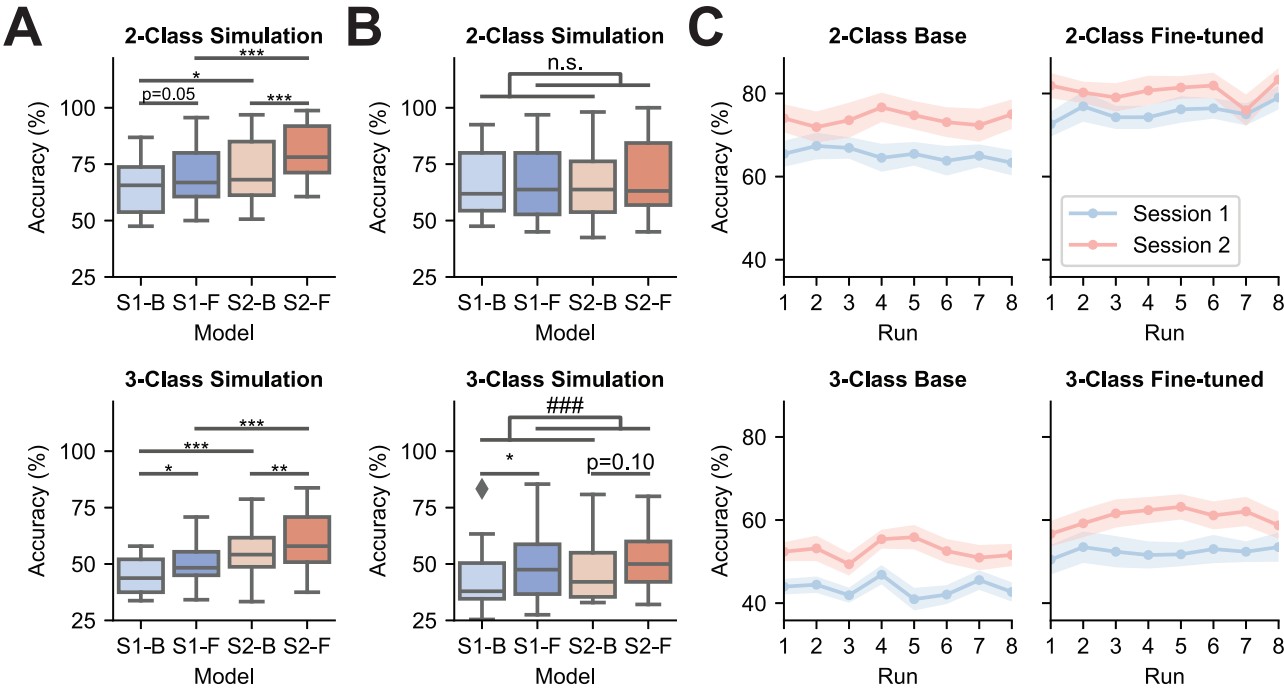

**Fig. 2 | Machine and human learning effects for MI-based robotic control tasks.** **A** Offline simulation results ($n = 21$ subjects) for Session 2 fine-tuned runs using Session 1 base models (S1-B), Session 1 fine-tuned models (S1-F), and Session 2 base models (S2-B), as well as the online decoding results for Session 2 fine-tuned runs (S2-F). The center lines indicate the median values. The boxes extend from the lower quartile to the upper quartile. Diamonds indicate outliers that are more than 1.5 times the interquartile range above the third quartile or below the first quartile. Comparisons between the sessions and the models were made using a two-tailed Wilcoxon signed-rank test with Bonferroni multi-comparison correction (*** if $p < 0.001$, ** if $p < 0.01$, * if $p < 0.05$, n.s. if no statistical significance is found). P-values: 0.0003 (S2-B vs. S2-F, 2-class), 0.0254 (S1-B vs. S2-B, 2-class), 0.0003 (S1-F vs. S2-F, 2-class), 0.0449 (S1-B vs. S1-F, 3-class), 0.0074 (S2-B vs. S2-F, 3-class), 0.0009 (S1-B vs. S2-B, 3-class), 0.0002 (S1-F vs. S2-F, 3-class). **B** Offline FBCSP decoding results on MI robotic finger control data ($n = 21$ subjects). S1-B, S1-F, S2-B, and S2-F denote decoding results for Session 1 base runs, Session 1 fine-tuned runs, Session 2 base runs, and Session 2 fine-tuned runs respectively. The center lines indicate the median values. The boxes extend from the lower quartile to the upper quartile. Diamonds indicate outliers that are more than 1.5 times the interquartile range above the third quartile or below the first quartile. Statistical analysis was conducted using a two-way repeated-measures ANOVA with main effects of session and model. Main effect of the model: ### if $p < 0.001$, n.s. if no statistical significance is found. Post hoc comparisons between models within the same session were made using a two-tailed Wilcoxon signed-rank test with Bonferroni correction (*** if $p < 0.001$, ** if $p < 0.01$, * if $p < 0.05$, n.s. if no statistical significance is found). P-values: 0.0005 (model, 3-class), 0.0114 (S1, 3-class). **C** Group-averaged performance trends across eight consecutive online runs within the same paradigm. The center lines indicate the mean values. The shaded areas represent the standard errors.

As with the MI paradigm, offline simulation results using the FBCSP-LDA model to decode EEG data from online sessions showed within-session performance improvements during the first 3-class session, but not under other conditions (Supplementary Fig. S7B). A significant session effect in the 3-class simulation results further suggested human learning involvement in the 3-class paradigm (Supplementary Fig. S7B). However, no consistent improvement in online decoding accuracy was observed across eight continuous runs within the same paradigm (Supplementary Fig. S7C).

Fine-tuning effectively mitigated the imbalances in the classification of different fingers by improving recall and precision for the most challenging classes (Fig. 3D, Supplementary Fig. S5B–D). With the fine-tuned model, 2-finger online ME robotic control was achieved with an average accuracy of 81.10% across 21 subjects, while the average accuracy for 3-finger control was 60.11%. Electrophysiological analysis showed a pronounced desynchronization in the alpha band power within the left sensorimotor region during right-hand finger movements, consistent with established neurophysiological patterns of motor planning and execution (Fig. 3E). Deep learning decoders similarly focused predominantly on the motor cortex (Fig. 3F, Supplementary Fig. S3B, S4D), aligning with known neurophysiological mechanisms.

**Effectiveness of fine-tuning persists through continued training**

Whereas significant improvement in online control performance across sessions has been observed in arm-level MI-based BCIs[11,55,56], to thoroughly investigate the learning effects of online MI robotic control at the individual finger level, 16 out of the 21 subjects completed a total of five MI online sessions. The base model was updated on a session-by-session base by training on all the available subject-specific data from previous MI sessions. The same metric, i.e. majority voting accuracy, was used for performance evaluation.

Two-way repeated-measures ANOVAs were conducted to evaluate the effects of session, model, and session-model interaction on performance. The results revealed significant learning effects across sessions for both 2-class and 3-class tasks (2-class: $F = 7.127$, $p < 0.001$; 3class: $F = 14.406$, $p < 0.001$, Fig. 4A, B), demonstrating noticeable performance improvements during online training. Additionally, a significant model effect indicated that fine-tuning consistently enhanced decoding accuracy as training sessions progressed (2-class: $F = 35.606$, $p < 0.001$; 3class: $F = 39.138$, $p < 0.001$, Fig. 4A, B). However, the session-model interaction effect was not significant, suggesting comparable learning speeds between the base and fine-tuned models (2-class: $F = 1.375$, $p = 0.253$; 3class: $F = 1.686$, $p = 0.164$).

To further investigate performance dynamics, comparisons between consecutive sessions were conducted using one-sided Wilcoxon signed-rank tests. These analyses revealed a noticeable boost in performance during the early stages of training, specifically between Session 2 and Session 1 (Fig. 4C, D). However, subsequent sessions showed no significant performance enhancements, despite numerical improvements (Fig. 4C, D). Specifically, one session of online training

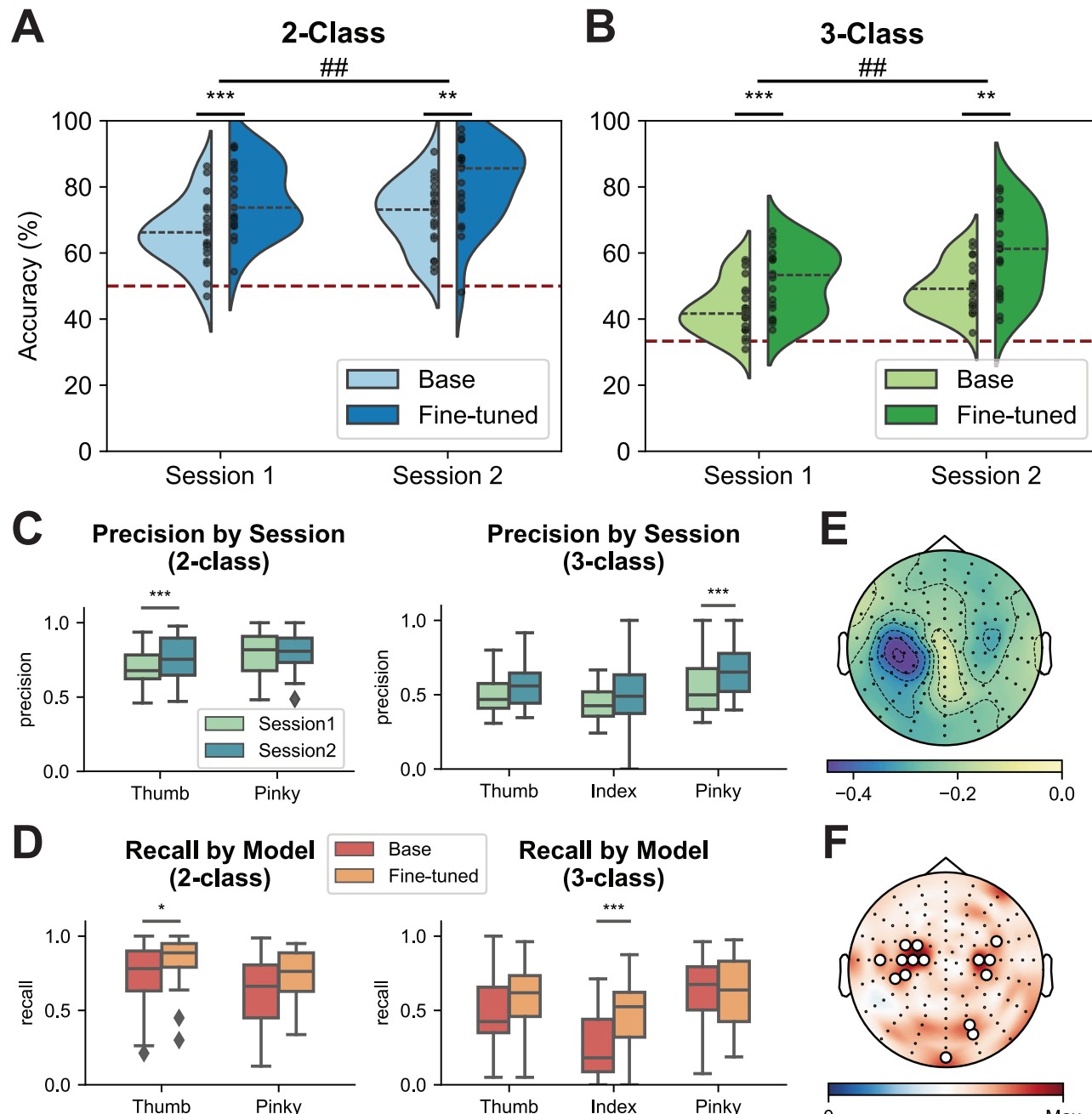

**Fig. 3 | Online performance in movement execution (ME) robotic finger control.**
Group-level online performance for 2-finger (**A**) and 3-finger (**B**) ME across sessions
($n = 21$ subjects). Comparisons between the base and fine-tuned models for Sessions 1 and 2 were conducted using a two-tailed Wilcoxon signed-rank test with Bonferroni correction for multiple comparisons (*** if $p < 0.001$, ** if $p < 0.01$). Two-way repeated-measures ANOVAs were applied with main effects of session and model, and comparisons between Sessions 1 and 2 were indicated by the main effect of session (## if $p < 0.01$). The red dashed line indicates the chance level, while the gray dashed line represents the median performance under each condition. The black dots represent every instance. P-values: 0.0001 (**A**, Session 1), 0.0055 (**A**, Session 2), 0.0075 (**A**, Session 1 vs. 2); 0.0003 (**B**, Session 1), 0.0026 (**B**, Session 2), 0.0018 (**B**, Session 1 vs. 2). Group-level precision of 2-finger and 3-finger ME decoding across sessions (**C**) and group-level recall of 2-finger and 3-finger ME decoding results for different classifiers (**D**) ($n = 21$ subjects). The center lines indicate the median values. The boxes extend from the lower quartile to the upper quartile. Diamonds indicate outliers that are more than 1.5 times the inter-quartile range above the third quartile or below the first quartile. Comparisons between the precision across sessions and recall across models were conducted using a two-tailed Wilcoxon signed-rank test with Bonferroni correction for multiple comparisons (*** if $p < 0.001$, * if $p < 0.05$). P-values: 0.0003 (**C**, Thumb, 2-class), 1.79e−05 (**C**, Pinky, 3-class), 0.0310 (**D**, Thumb, 2-class), 3.35e−05 (**D**, Index, 3-class). **E** Group averaged alpha band (8–13 Hz) ERD topology for ME online control tasks. **F** Group-averaged saliency topological maps of the EEGNet models for online ME sessions. Channels with the highest 25% saliency values are highlighted by white circles on the electrodes.

was sufficient for the EEGNet models to extract distinguishable features from scalp EEG signals, performing comparably to later training sessions. This highlights the efficiency of the training process. To evaluate whether the compact architecture of EEGNet-8,2, with only a few thousand trainable parameters, limited its capacity to leverage larger datasets, we performed offline analyses using a wider and deeper variant (deepEEGNet). This model increased the number of filters in each layer and incorporated two additional separable convolutional

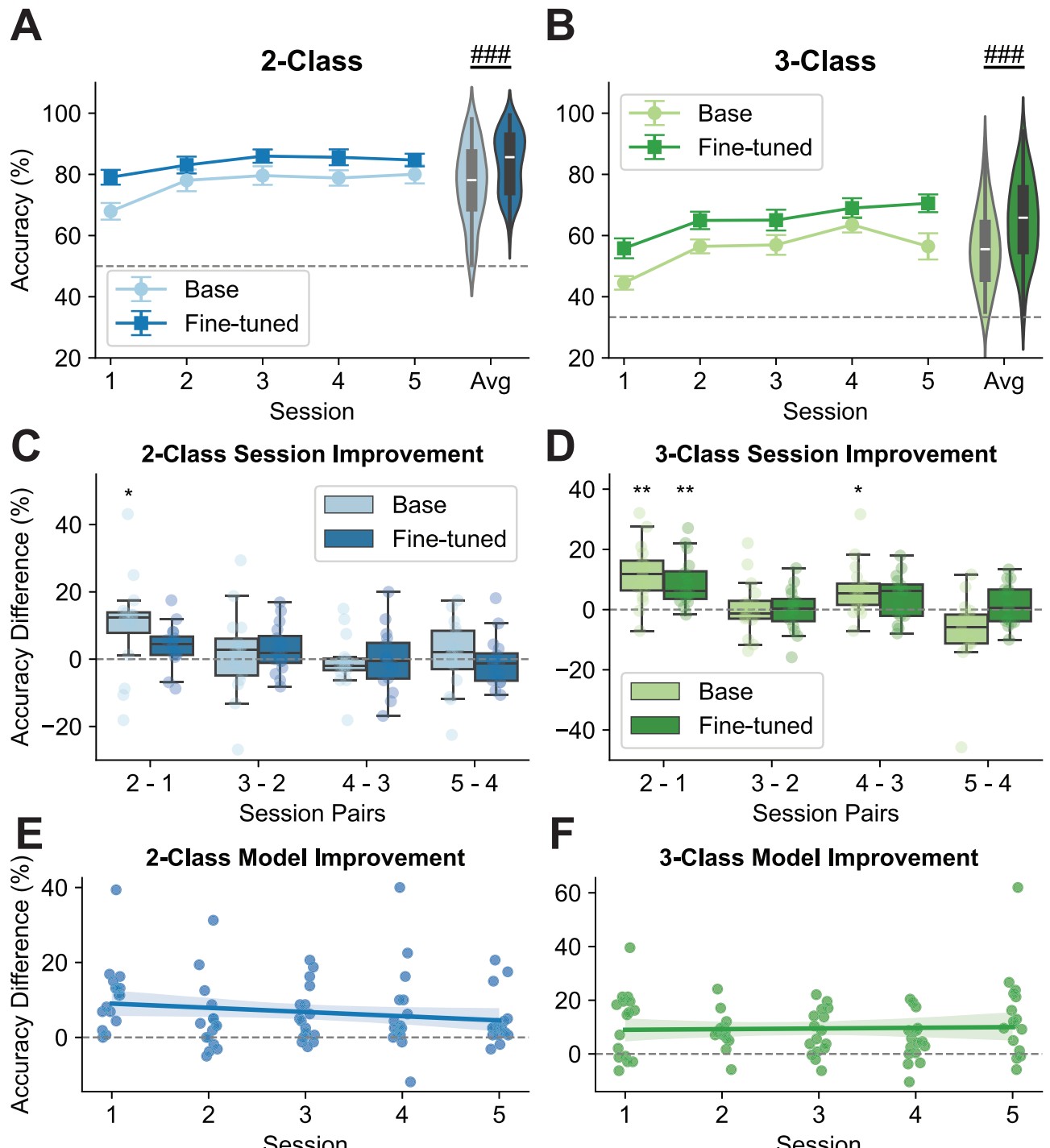

**Fig. 4 | Effects of online training on MI robotic finger control.** Group-level trends in 2-finger (**A**) and 3-finger (**B**) MI accuracy over five online sessions ($n = 16$ subjects). The center lines indicate the mean values. The gray dashed line represents the chance level, and error bars indicate the standard errors. The performance distribution for the Base and Fine-tuned models across all sessions is shown on the right. The center lines indicate the median value. The boxes extend from the lower quartile to the upper quartile, and the lines indicate 1.5 times the interquartile range. Statistical analysis was conducted using a two-way repeated-measures ANOVA with main effects of session and model. Main effect of the model: ### if $p < 0.001$. P-values: 2.6e−05 (**A**), 1.53e−05 (**B**). Pairwise comparisons of 2-finger (**C**) and 3-finger (**D**) MI accuracy between consecutive sessions

($n = 16$ subjects). The center lines indicate the median values. The boxes extend from the lower quartile to the upper quartile. The whiskers span up to 1.5 times the interquartile range. One-sided Wilcoxon signed-rank tests were performed for each session pair and model type. Statistical significance after Bonferroni correction is denoted as ***$p < 0.001$, **$p < 0.01$, *$p < 0.05$. P-values: 0.0260 (**C**, 2-1, Base), 0.0024 (**D**, 2-1, Base), 0.0012 (**D**, 2-1, Fine-tuned), 0.0103 (**D**, 4-3, Base). Linear regression analysis of the improvement in 2-finger (**E**) and 3-finger (**F**) MI accuracy achieved by the fine-tuned model compared to the Base model across five sessions. The center line indicates the linear regression model fit of the data. The shaded region indicates the 95% confidence interval for the regression.

layers before the flatten layer (Supplementary Table S1). Despite modest numerical improvements in decoding accuracy (2-class: +1.21%; 3-class: +1.52%) compared to the simulated results using EEGNet, the overall learning trend remained consistent using deepEEGNet (Supplementary Fig. S8). For this comparison, deepEEGNet results were benchmarked against EEGNet's simulated outcomes rather than online performance metrics to mitigate potential confounding effects of real-time feedback. This persistent plateau suggests that the observed trend may stem from inherent limitations in EEG-derived features for sophisticated finger movement decoding, rather than model capacity alone.

To quantify the effects of fine-tuning, linear regression analyses were performed on the accuracy differences between fine-tuned models and base models across training sessions. The results demonstrated a significant superiority of fine-tuned models as training progressed, reflected by significant positive offsets (2-class: $t = 4.251$, $p < 0.001$; 3class: $t = 2.992$, $p = 0.004$, Fig. 4E, F). However, the session effect was not significant (2-class: $t = -1.558$, $p = 0.311$; 3class: $t = 0.286$, $p = 0.776$, Fig. 4E, F), indicating that the decoding capability of the fine-tuned models remained consistently superior to that of the base models throughout training. These findings confirm the effectiveness of fine-tuning in mitigating the cross-session variability and improving the performance of deep-learning models across multiple sessions.

In addition to online behavioral results, offline decoding and electrophysiological analysis were conducted to examine the training effects on the human side. Offline decoding of single-session EEG data using an FBCSP-LDA model revealed no improvement in signal discriminability for binary tasks (Supplementary Fig. S9A). However, for ternary tasks, significant improvements were observed in fine-tuned runs compared to base runs, as indicated by a two-way repeated-measures ANOVA ($F = 15.028$, $p = 0.001$, Supplementary Fig. S9B), and the enhancement in the data discriminability mainly occurs in the first session (Cohen's $d = 0.502$ for Session 1, Supplementary Fig. S9B). This indicates a plateau in human MI learning at the early stages of training. One-way ANOVA analysis of the EEG data collected during the online training sessions showed that ERD levels at channel C3, which is believed to produce the strongest neural signals related to right-hand MI, remained consistent throughout five training sessions (2-class: $F = 0.432$, $p = 0.784$; 3-class: $F = 0.284$, $p = 0.887$, Supplementary Fig. S10). These findings suggest that the proposed paradigm requires minimal training to achieve satisfactory performance.

## Online smoothing stabilizes the robotic control

Despite the strong performances achieved by deep-learning-based decoders in finger-level robotic control tasks, this sophisticated control paradigm remains challenging. The primary difficulties stem from the low signal-to-noise ratio and limited spatial resolution of scalp EEG, particularly in ternary classification tasks with an increased number of categories[57]. To further enhance robotic control performance, we developed an online smoothing approach aimed at stabilizing robotic movements by incorporating previous decoding outputs into the decision-making process (Supplementary Movies S3, S4). Out of the 21 subjects studied, 16 participated in one ME online session and one MI online session, with each session comprising four runs using the original control mechanism and four runs using the smoothing mechanism. In addition to accuracy, we evaluated the efficacy of the smoothing approach using two additional metrics: Label Shift, defined as the number of changes in the continuously predicted results within one trial, and All-Hit Ratio, calculated as the percentage of trials in which all predictions matched the true label.

Separate linear mixed-effect models were constructed for each metric across the 2-finger ME paradigm (Fig. 5A), 3-finger ME paradigm (Fig. 5B), 2-finger MI paradigm (Fig. 5C), and 3-finger MI paradigm (Fig. 5D). No significant differences were observed between the smoothed outputs and the original ones in terms of accuracy, which

suggests that the smoothing approach maintained task performance across these paradigms. Furthermore, in all four paradigms, we observed a significant reduction in the number of label shifts, reflecting enhanced stability in the output control commands. Comparing the percentage of All-Hit trials before and after applying the smoothing approach, we found that it effectively filtered out small perturbations during the tasks, leading to better overall control (Fig. 5E, F). These findings underscore the potential of the smoothing approach to improve the robustness and reliability of EEG-based robotic control systems.

## Offline decoding between different finger pairs

The decoding results from offline sessions provided valuable insights into the discriminability of different finger groups, guiding the design of online experimental paradigms. Subjects were instructed to perform either ME or MI of the thumb, index finger, middle finger, and pinky of their dominant hand (right hand). Offline decoding was performed on all possible 2-finger, 3-finger, and 4-finger combinations using EEGNet and FBCSP.

Among all the 2-finger groups, the thumb–pinky pair demonstrated the highest decoding performance across paradigms and classifiers, achieving an average accuracy of 77.58% for the MI task and 75.65% for the ME task using EEGNet. Conversely, the index finger and the middle finger combination exhibited the least discriminability (Fig. 6A, B). To explore the electrophysiological basis for the variation in discriminability across finger pairs, the distances between EEG channels showing maximum ERD activation for each pair were computed and compared. The thumb–pinky and index–pinky pairs exhibited the greatest distance in MI-induced ERD activation within the alpha band, aligning with the offline decoding results and somatotopic mapping (mean distance: 46.316 mm for thumb–pinky and 42.262 mm for index–pinky, Supplementary Fig. S11). For 3-finger combinations, the thumb-index-pinky and thumb-middle-pinky groups showed the highest accuracy, as reflected in the averaged results. In the 4-finger classification, an average accuracy of 43.61% for the MI task and 42.17% for the ME task was obtained. These offline decoding results demonstrate the enhanced performance of EEGNet in finger-level EEG-BCI decoding compared to non-deep-learning decoders like FBCSP, underscoring the potential of deep-learning-based methods in accurately distinguishing between various finger movements or motor intentions.

## Frequency-specific contributions to finger discrimination

The suppression of oscillatory activities within the alpha (8–13 Hz) and beta (13–30 Hz) frequency bands in the contralateral sensorimotor region during both ME and MI tasks has been widely discussed[58,59]. Previous studies have shown that alpha and beta frequency components offer effective discriminations between limb-level MI tasks[10,11,18,60]. To explore the frequency-specific information encoded in EEG during finger ME and MI tasks, we extracted alpha and beta frequency components from offline EEG recordings. Task-specific ERD topologies revealed highly similar spatial patterns in both alpha and beta bands when subjects performed or imagined movements of different fingers (Fig. 6C, D). During ME tasks, a bilateral power decrease in both frequency bands was observed over the sensorimotor region, with the strongest suppression on the contralateral side. In contrast, MI tasks exhibited a more localized ERD in the left sensorimotor region across both frequency bands. When comparing the modulation of EEG oscillations induced by ME and MI tasks, we observed stronger bilateral desynchronization during ME, consistent with previous findings[59,61–63]. Additionally, power modulations in the alpha and beta bands were also observed in parietal regions.

Since the analysis focused on the BCI responders after two rounds of screening based on their ME and MI performance, which might influence the group-level alpha and beta band ERD trends, we

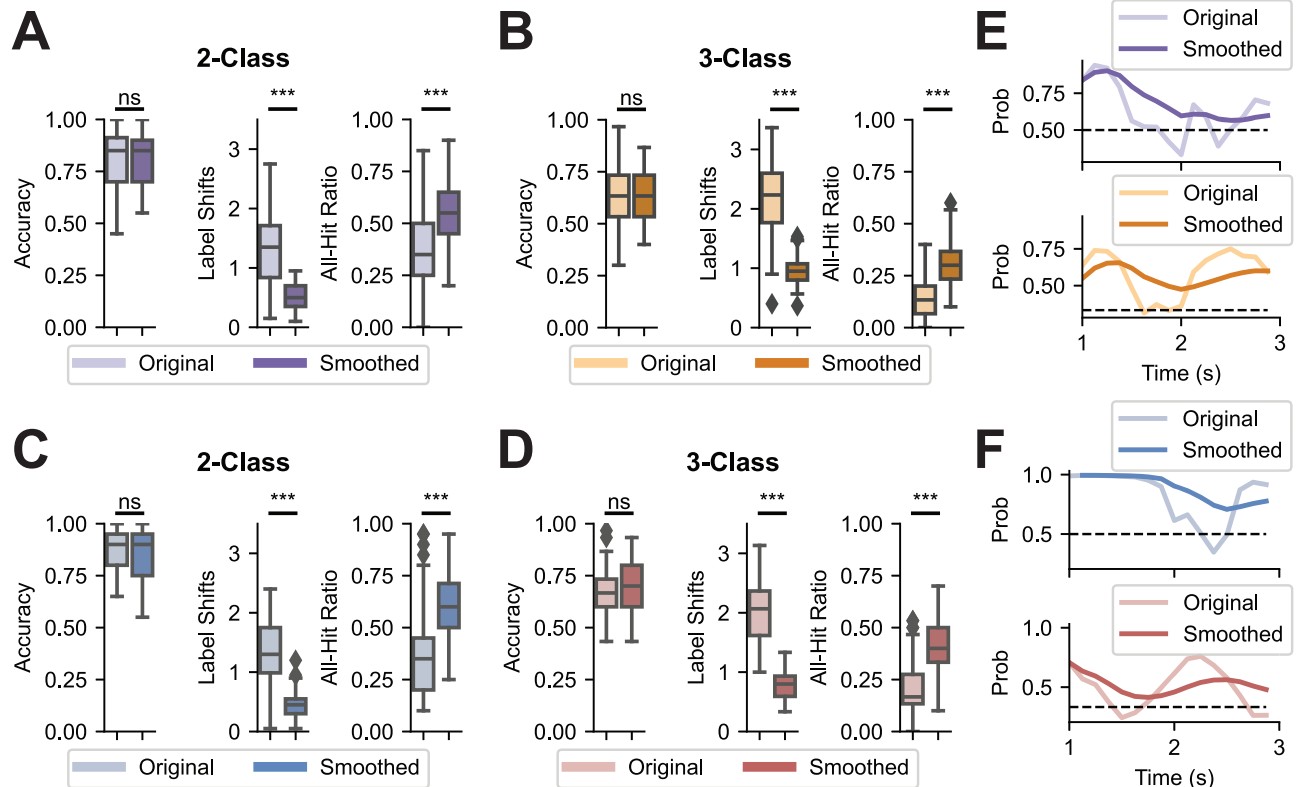

**Fig. 5 | Effects of online smoothing.** Comparison between the original outputs from the fine-tuned EEGNet model and the smoothed outputs for 2-finger ME (**A**), 3-finger ME (**B**), 2-finger MI (**C**), and 3-finger MI (**D**) online tasks (n = 16 subjects). Accuracy is calculated using the majority vote method; Label Shifts represents the average number of shifts in predicted labels within one single trial, which quantifies how stable the decoder is; All-Hit Ratio is defined as the percentage of the trials that are classified correctly throughout the whole trial. The center lines indicate the median values. The boxes extend from the lower quartile to the upper quartile. Diamonds indicate outliers that are more than 1.5 times the interquartile range above the third quartile or below the first quartile. Comparisons between the two conditions were made using a linear mixed effects model (*** if $p < 0.001$, ** if $p < 0.01$, * if $p < 0.05$). No $p$-value adjustments were made. *P*-values: 7.66e−13 (**A**, Label Shifts), 7.97e−06 (**A**, All-Hit Ratio), 2.16e−28 (**B**, Label Shifts), 2.78e−10 (**B**, All-Hit Ratio), 3.43e−12 (**C**, Label Shifts), 9.04e−07 (**C**, All-Hit Ratio), 3.49e−33 (**D**, Label Shifts), 9.77e−16 (**D**, All-Hit Ratio). Comparison between the original and the smoothed probabilities for the target class of one example 2-finger trial (top) and one example 3-finger trial (bottom) for ME (**E**) and MI (**F**) tasks. The black dashed line represents the probability at the chance level.

conducted analyses comparing ERD activations between qualified participants and non-responders excluded for unsatisfactory performance. Among participants excluded for poor ME performance (n = 13), alpha and beta ERD in the contralateral sensorimotor region remained detectable but markedly weaker than in responders (Supplementary Fig. S12A, B, F). For individuals excluded due to inadequate MI performance despite intact ME ability (n = 5), contralateral ERD was evident during ME tasks but absent during MI (Supplementary Fig. S12C, D). Across both responders and non-responders, MI tasks consistently elicited weaker alpha and beta ERD than ME tasks (Supplementary Fig. S12E), supported by medium-to-large effect sizes across groups. Non-responders also exhibited reduced ERD magnitudes compared to responders in both ME and MI tasks (Supplementary Fig. S12F), indicating their inability to generate robust sensorimotor cortical activation. These findings demonstrate that the attenuated ERD during MI relative to ME in alpha and beta bands is a consistent feature across both groups.

Beyond the alpha and beta rhythms, movement-related cortical potentials (MRCPs) in the low-frequency band (0.3–3 Hz) have been frequently used to differentiate hand and finger movements[39,64,65]. To assess the contribution of these slow components, we visualized time-resolved amplitude topoplots and MRCP waveforms at the selected channel (C3) (Supplementary Fig. S13). All four fingers elicited significant changes in low-frequency EEG amplitudes within the first 1.5 seconds following trial onset (Supplementary Fig. S13A, B). The amplitude modulation began in the contralateral fronto-central

regions and spread to the contralateral parietal areas over time, in both ME and MI conditions. No significant changes were observed prior to the trial onset or after 1.5 seconds. Compared to MI, ME elicited stronger amplitude changes. MRCPs were present in both finger ME and MI, characterized by a negative deflection in the electrode shortly after trial onset (Supplementary Fig. S13C, D). Although finger-specific differences were observed during a brief window around the MRCP deflection, low-frequency activity across individual fingers showed considerable overlap throughout the trial.

To assess the discriminability of individual finger movements across frequency bands, we filtered the EEG data into delta (0.5–4 Hz), theta (4–8 Hz), alpha, and beta bands. Each was separately input into EEGNet, and the resulting decoding accuracies were compared with those obtained from broadband (4–40 Hz) data. Decoding performance significantly declined when using individual frequency bands compared to the 4–40 Hz input, highlighting the contribution of broadband information in decoding finger movements (Fig. 6E, F). Among the single-band conditions, the alpha band yielded significantly higher accuracy than the delta, theta, and beta bands, while the beta band significantly outperformed the delta band. To further examine the role of low-frequency components, we expanded the broadband range to 0.3–40 Hz to include delta activity. The decoding performance of this extended broadband input remained superior to that of individual bands, and was comparable to that of the alpha band (Supplementary Fig. S14). These findings highlight the pivotal role of broadband EEG signals and specifically underscore the critical role of

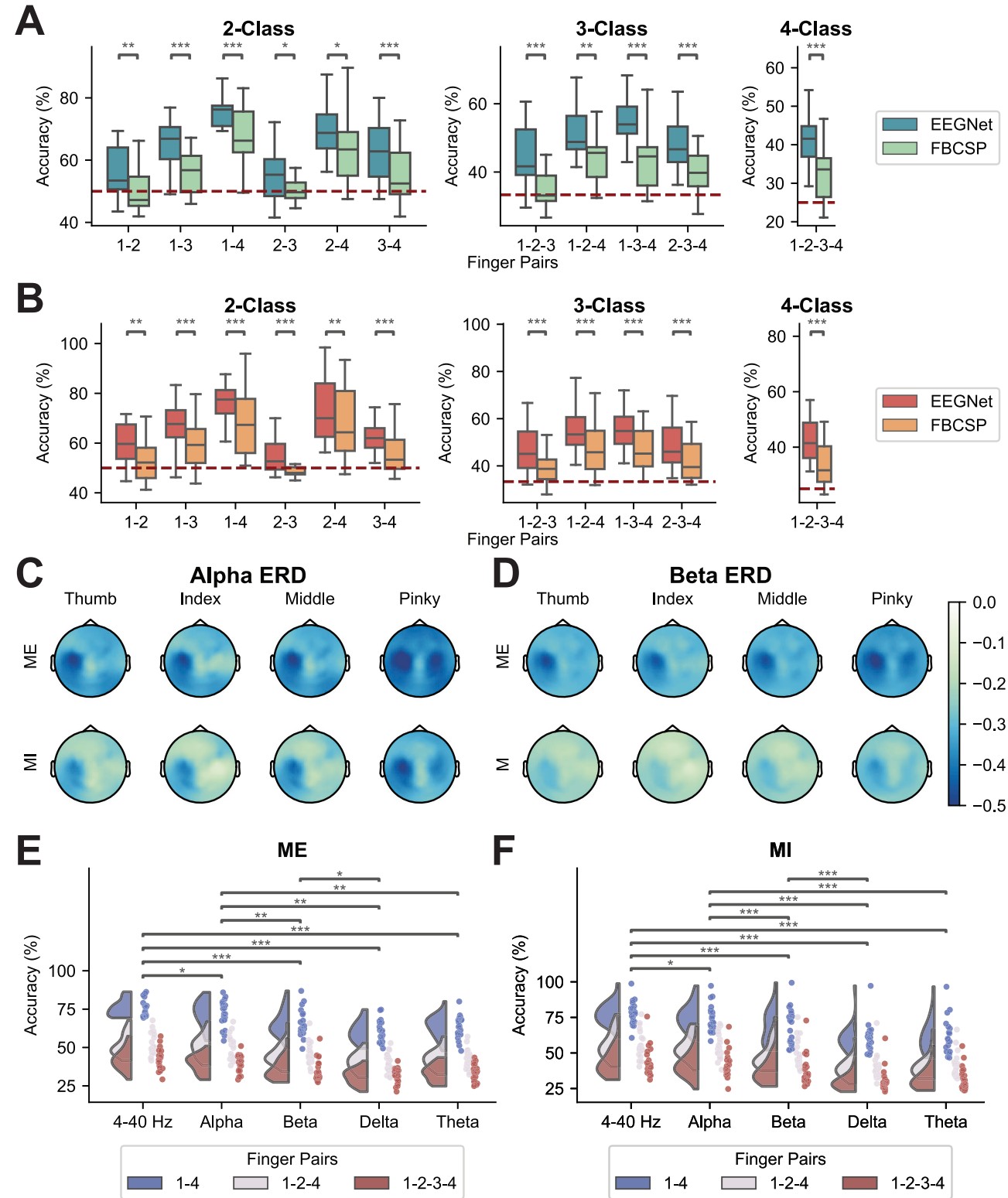

alpha and beta oscillations in distinguishing individual finger movements.

## Discussion

In this study, we demonstrate continuous robotic finger control using MI and ME through deep learning techniques (Fig. 7A, Supplementary Movies S1–S4). Our findings highlight the separability of scalp EEG signals corresponding to different finger movements or movement intentions of the same hand, decoded using a deep-learning-based model. Significant improvements in task performance were achieved through online training and model fine-tuning. The online paradigms included binary tasks (thumb and pinky motions) and ternary tasks (thumb, index finger, and pinky motions). In a group of 21 able-bodied experienced BCI users, we achieved average accuracies of 80.56% for 2-finger MI control and 60.61% for 3-finger MI control after a single session of online training and fine-tuning, demonstrating the system's effectiveness. Further online training for MI tasks revealed that one session was sufficient to attain satisfactory performance in the studied

**Fig. 6 | Offline decoding performance and electrophysiological analysis.** Group-level offline decoding performance of finger ME (**A**) and MI (**B**) using EEGNet and FBCSP ($n$ = 21 subjects), where 1 represents thumb, 2 represents index finger, 3 represents middle finger, and 4 refers to pinky. The red dashed lines indicate the chance levels. The center lines indicate the median values. The boxes extend from the lower quartile to the upper quartile. The whiskers span up to 1.5 times the interquartile range. Comparisons between two decoders were conducted using a two-tailed Wilcoxon signed-rank test with Bonferroni correction for multiple comparisons. *P*-values: 0.0011 (**A**, 1-2), 0.0003 (**A**, 1-3), 0.0006 (**A**, 1-4), 0.0132 (**A**, 2-3), 0.0308 (**A**, 2-4), 0.0009 (**A**, 3-4), 3.81e−06 (**A**, 1-2-3), 0.0011 (**A**, 1-2-4), 0.0002 (**A**, 1-3-4), 2.67e-05 (**A**, 2-3-4), 3.81e-06 (**A**, 1-2-3-4), 0.0054 (**B**, 1-2), 0.0001 (**B**, 1-3), 5.72e−06 (**B**, 1-4), 0.0009 (**B**, 2-3), 0.0011 (**B**, 2-4), 1.14e−05 (**B**, 3-4), 0.0003 (**B**, 1-2-3), 0.0002 (**B**, 1-2-4), 1.14e-05 (**B**, 1-3-4), 0.0005 (**B**, 2-3-4), 0.0002 (**B**, 1-2-3-4). Group-level task-specific alpha band (8–13 Hz) (**C**) and beta band (13–30 Hz) (**D**) ERD topographies ($n$ = 21 subjects). From left to right, ERD topographies corresponding to thumb, index, middle, and pinky movements are displayed. The top row presents results from ME data, while the bottom row presents results from MI

data. Comparison of offline EEGNet decoding performance ($n$ = 21 subjects) for ME (**E**) and MI (**F**) tasks using EEG signals filtered with different bandpass settings (4–40 Hz, alpha band, beta band, delta band, theta band). The x-axis shows classification results for different frequency bands. Offline classifications were performed on thumb vs. pinky (1-4), thumb vs. index finger vs. pinky (1-2-4), and all four fingers (1-2-3-4). Two-way ANOVA was conducted across different frequency bands and finger pairs, and statistical significance was observed in all task conditions. Significance stars indicate post hoc pairwise comparison results using an FDR-corrected two-tailed Wilcoxon signed-rank test (*** if $p < 0.001$, ** if $p < 0.01$, * if $p < 0.05$). *P*-values: 0.0181 (**E**, 4–40 Hz vs. Alpha), 9.53e−06 (**E**, 4–40 Hz vs. Beta), 4.45e−05 (**E**, 4–40 Hz vs. Delta), 4.45e−05 (**E**, 4–40 Hz vs. Theta), 0.0018 (**E**, Alpha vs. Beta), 0.0020 (**E**, Alpha vs. Delta), 0.0023 (**E**, Alpha vs. Theta), 0.0197 (**E**, Beta vs. Delta), 0.0329 (**F**, 4–40 Hz vs. Alpha), 3.17e−06 (**F**, 4–40 Hz vs. Beta), 3.17e−06 (**F**, 4–40 Hz vs. Delta), 1.19e−05 (**F**, 4–40 Hz vs. Theta), 0.0002 (**F**, Alpha vs. Beta), 3.17e−06 (**F**, Alpha vs. Delta), 4.76e−05 (**F**, Alpha vs. Theta), 8.74e−05 (**F**, Beta vs. Delta).

finger control tasks. This finding underscores the practicality of our system for real-world applications, as it requires minimal training. Additionally, we developed an online smoothing approach to enhance control stability by mitigating noise-induced fluctuations. The system's performance demonstrates its ability to translate brain signals into precise motor commands for the same hand through intuitive mapping, a critical feature for real-world BCI-based robotic control applications.

Intracortical BCI systems have made significant strides in robotic device control by directly interfacing with neural activity, achieving high precision and real-time feedback for multi-degree-of-freedom tasks. Studies have demonstrated high-performance neuroprosthetic control[7–9], simultaneous, independent finger movement control[31,33], and BCI systems for quadcopter control and typing based on finger decoding[34,35]. High-density recordings enable intuitive decoding of complex limb motions, while the low spatial resolution of EEG limits its task complexity despite its noninvasiveness and portability. Therefore, most state-of-the-art EEG-BCI systems employ motor imagery tasks unrelated to the intended output actions the user aims to achieve[10,11,47]. The discrepancy between human thoughts and the control commands requires practice and training before achieving satisfactory performance. This has led to research efforts aimed at decoding upper extremity movements or intentions reflective of desired actions, including elbow, forearm, and hand movements[65,66], grasping types[67,68], and finger-related tasks[36–38]. While some studies have translated dexterous grasping into online control using scalp EEG[67], existing research on finger-level movement decoding has been limited to offline analysis. Our work extends previous explorations in sophisticated upper limb movement decoding and explores the possibility of real-time, naturalistic robotic control at the individual finger level with both movement execution and motor imagination.

Compared to previous studies utilizing invasively acquired signals, our study demonstrated a comparable response time with a latency of 1 s. In contrast, a recent study on an ECoG-based finger decoding system reported a group delay of ~1.17 s[30], while another study using implanted electrode arrays for finger decoding introduced a delay period of 1.5 s before providing feedback[33]. We observed a continuous increase in the decoding probability of the target finger within the first second after the task onset in both 2-finger and 3-finger control (Supplementary Fig. S15). Following the onset of the feedback period, the decoding probability of the target finger remained elevated and consistently high until the trial ended, indicating that reliable control can be achieved with minimal delay. Furthermore, we achieved satisfying online decoding performance, with an average decoding accuracy of 80.56% for 2-finger MI control, significantly surpassing the offline movement decoding accuracy for the same finger group (thumb vs. pinky) reported in earlier research[38]. Though our study

does not yet attain the level of independent and simultaneous online finger control achieved by intracortical BCIs[31,32], it represents a significant advancement in the field of noninvasive BCI in dexterous robotic control.

Our findings demonstrate that deep-learning strategies can effectively resolve subtle motor-related EEG patterns obscured by the volume conduction effects[42], enabling precise real-time BCI control. The subtle differences in electrical activity caused by individual finger movements or imagined movements are challenging to detect with conventional BCI decoders due to signal smearing. The present findings demonstrate that deep learning enables efficient and automated feature extraction from EEG time series, significantly reducing the time required for signal processing compared to traditional machine-learning-based methods. This approach facilitates the online classification of minimally processed short EEG segments, allowing for real-time feedback in an online setting and ultimately improving BCI performance. Furthermore, the proposed fine-tuning paradigm enhanced BCI performance by mitigating session-to-session variability, a challenge that standard pre-trained deep learning models were unable to address.

The improvement in online task performance across sessions and through fine-tuning underscores the critical role of online training in enhancing MI and ME control (Figs. 1A, B, 3A, B). The superior offline decoding performances achieved using Session 2 models, as compared to Session 1 models reveal evidence of active machine learning facilitated by increased training data (Fig. 2A, Supplementary Fig. S7A). Additionally, we demonstrated that fine-tuning significantly enhances decoding performance even with limited training data (Figs. 1A, B, 2A, 3A, B, Supplementary Fig. S7A). The within-session comparison of performances highlights fine-tuning as a critical strategy for mitigating inter-session variability in EEG signals. Beyond machine learning effects, we observed human learning across sessions and within the first online training session for ternary MI and ME tasks (Fig. 2B, Supplementary Fig. S7B). However, the binary paradigms exhibited no significant human learning effect, potentially due to variations in cognitive load and task complexity. The cognitive load theory suggests that balancing task difficulty is crucial for effective learning, while overly simple tasks fail to sufficiently stimulate cognitive processing[69,70]. These findings offer valuable insights into designing experimental paradigms optimized for human learning.

Our results underscore the critical role of the contralateral sensorimotor region in differentiating executed and imagined finger movements, aligning with neurophysiological patterns of motor intention and execution (Figs. 1E, 3E, Supplementary Fig. S4). Notably, however, decoding performance declined when restricted to the contralateral hand knob region alone (Supplementary Fig. S4B, C), suggesting the potential contributions of other coactivated regions

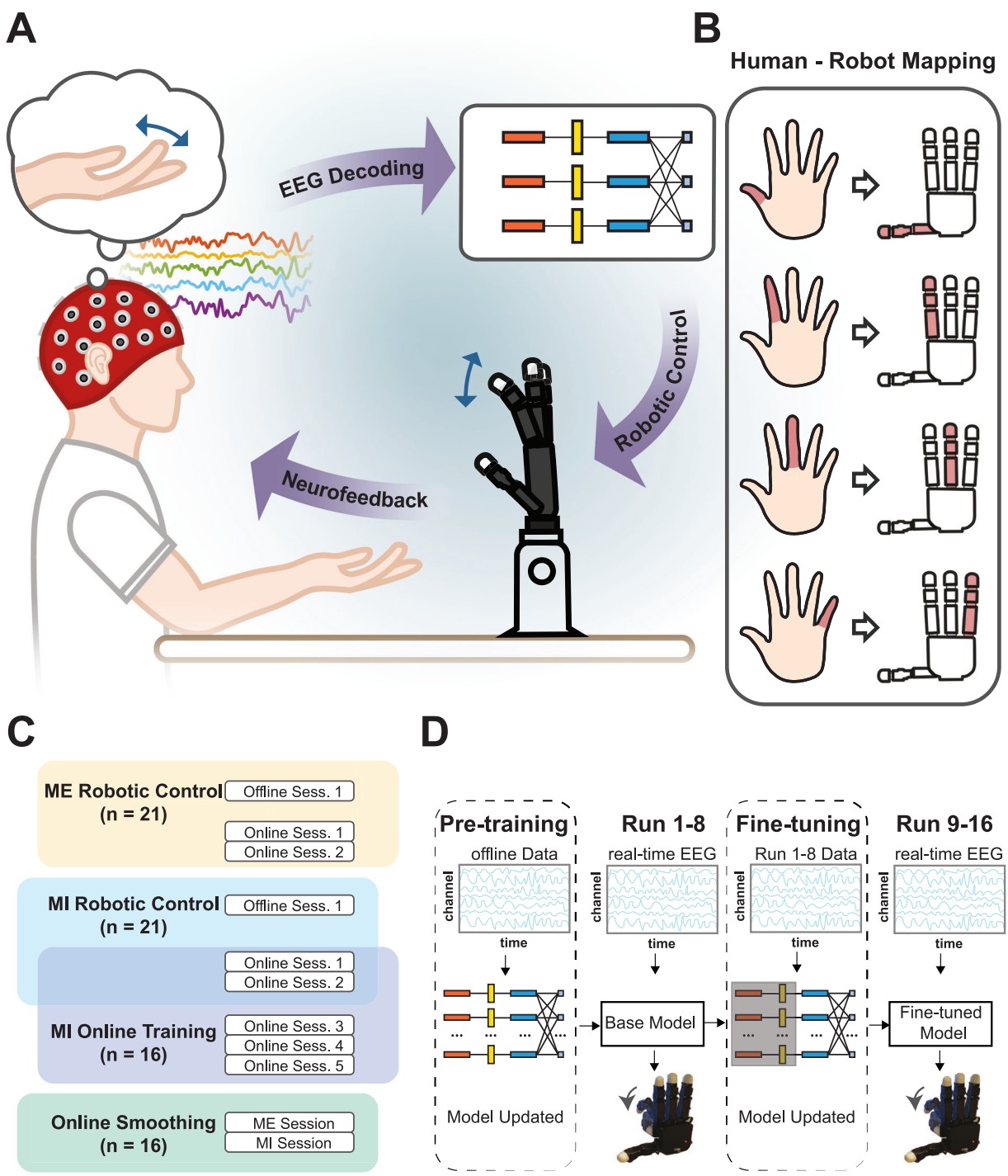

**Fig. 7 | Experimental paradigm. A** Experimental Setup. Motor imagery or execution of single fingers was used to drive the robotic finger motion. **B** Mapping from human motions or imagination of motions of fingers (right hand) to the robotic finger control (right hand). **C** Overview of experimental sessions for each participant. There were four different phases of experiments with a varying number of participants involved depending on their availability. **D** Overview of the online session design. In the first half session, a pre-trained EEGNet model was utilized to provide real-time robotic hand control at finger level while subjects were performing the finger movement imagery or execution. The latter part of the session used an EEGNet model fine-tuned on the data from the first 8 runs.

during online training. These may include attentional modulation and bilateral sensorimotor coordination. The saliency topological maps highlight the importance of parietal and occipital channels in online decoding, suggesting a role for visual attention during MI and ME, as supported by offline EEG analysis (Figs. 1E, F, 3E, F). Some participants reported focusing on the target finger, either through direct gaze or peripheral vision, indicating visual engagement. The occipital lobe, where the primary visual cortex is located, is the primary recipient of

visual feedback. The parietal activations are known to reflect covert attention and oculomotor intention as part of the fronto-parietal network, which plays a crucial role in the top-down attention required for neurofeedback[71]. Supporting this observation, previous studies have reported activations in the parietal-occipital regions during tasks involving conscious visuospatial attention modulation[72-74]. In our study, ERD in the ipsilateral sensorimotor region was observed, particularly during finger ME, which may be attributed to the deactivation

of ipsilateral sensorimotor structures[60]. Various neural patterns within the ipsilateral sensorimotor region were reported and discussed across subjects and studies[58,63,75]. Our saliency maps and subregion analyses suggest the involvement of task-related EEG changes in the ipsilateral side for decoding finger movements (Figs. 1F, 3F, Supplementary Fig. S4D, E). In addition, simulations of low-density EEG systems with broad scalp coverage revealed stable decoding performance despite reduced channel counts, emphasizing the primacy of broad spatial coverage over high-density sampling (Supplementary Fig. S16). This finding reflects EEG's intrinsic limited spatial resolution due to volume conduction, which manifests focal cortical activity across scalp regions. Consequently, sparse but spatially distributed electrodes can effectively capture the distributed neural signatures necessary for decoding, reducing system complexity and computational load without sacrificing accuracy. In summary, while the sensorimotor cortex is central to finger movement decoding, whole-scalp coverage captures complementary neural processes that improve decoding robustness[44]. The minimal performance drop with lower-density systems underscores that practical EEG-based BCIs can prioritize broad spatial coverage over high channel density, which provides important guidance for designing real-world systems with reduced complexity and increased usability.

Interestingly, task performance for robotic finger control plateaued early in the online sessions, with additional training yielding no further improvement in accuracy (Fig. 4C, D). Similarly, C3 ERD values showed numerically stronger responses in the first three sessions for both the 2-class and 3-class tasks but plateaued after Session 3 (Supplementary Fig. S10). Moreover, offline decoding results using FBCSP and deepEEGNet revealed enhanced data discriminability primarily in the first session (Supplementary Fig. S8, S9), suggesting that performance saturation likely reflects inherent limitations in EEG-derived features for precise finger movement decoding, rather than constraints imposed by model capacity. This contrasts with other studies reporting continuous improvement through up to 10 sessions[55]. This discrepancy may be due to the intuitive mappings between dominant hand movements and output commands, reducing cognitive load and shortening the learning curve. Additionally, participants' prior experience with limb-level MI tasks may have facilitated faster adaptation. The observed early performance plateau suggests that extensive additional training may be unnecessary for experienced BCI users, even if they are new to finger-level MI. This finding has practical implications for designing efficient MI-based BCI systems. Future research should explore the effects of finger MI training on naïve subjects to determine if the rapid plateau is unique to experienced users or generalizable to those without previous MI experiences. Furthermore, despite the initial investigation on expanding the architecture size, systematical evaluation of the state-of-the-art decoders may potentially enhance decoding performance over extended training periods in future applications[76].

We observed comparable online performance between the ME-controlled and MI-controlled paradigms (Supplementary Fig. S17A, B). Both tasks demonstrated similar decoding patterns over time (Supplementary Fig. S15). The similarity in the underlying neural mechanisms between ME and MI has been extensively discussed, supporting our findings[77,78]. However, previous research has shown that greater primary motor cortex involvement was observed during ME compared to MI tasks[59,61,62], which is in alignment with our offline EEG analysis on the alpha and beta band ERD in both experienced BCI users and non-responders (Fig. 6D, Supplementary Figs. S12, S17C). The highly comparable performances observed between MI and ME-controlled online paradigms are likely due to enhanced engagement from real-time feedback, which helps optimize arousal levels according to the Yerkes-Dodson law, thereby enhancing cognitive and neural engagement[62,79,80]. Neurofeedback has been shown to amplify MI-related sensorimotor rhythms beyond levels seen in ME[62] and improve

classification accuracy[81]. We believe that the real-time feedback in our paradigm increased participants' arousal and attention, improving the specificity of MI-related neural activity and narrowing the decoding gap between MI and ME. Furthermore, decoding performance is not strictly determined by ERD magnitude but by the classifier's ability to extract task-relevant biomarkers, including subject-specific oscillatory patterns and whole-scalp distribution. EEGNet leverages subject- and task-specific narrow-band spectral features, despite a general focus on frequency components within alpha and beta ranges (Supplementary Fig. S18). Unlike the conventional ERD analysis, which relies on fixed wide-band power suppression, deep-learning-based decoders automatically identify subtle temporal patterns differentiating individual finger movements and imaginations. Such approaches can capture nuanced neural patterns that encode task-relevant information, even without pronounced ERD expression. Moreover, although contralateral sensorimotor cortex activity prominently contributes to motor decoding, incorporating whole-scalp data significantly improved classification accuracy compared to using motor-region-only signals (Supplementary Fig. S4B, C). Deep learning's capacity to identify distributed task-related features[44] likely compensates for weaker MI-induced ERD by integrating supplementary task-relevant physiological activities, including attentional modulation, bilateral sensorimotor coordination, and other motor planning processes. Together, these findings suggest that comparable performance between finger ME and MI arises from feedback-driven neural engagement and data-driven extraction of diverse physiological biomarkers.

Offline decoding results are presented for other 2-finger and 3-finger combinations as well as 4-finger classification. Among all the binary conditions, the thumb and pinky showed the best separability in both ME and MI tasks, while the decoders failed to distinguish between index and middle finger movements. This finding aligns with previous studies showing greater inter-digit distances for pairs of more distant digits and higher overlap in their activation maps for closer pairs[82]. This presents a challenge for scalp EEG to distinguish between neighboring fingers due to its limited spatial resolution. In the 3-class decoding, the similar performance achieved by the thumb-index-pinky group and the thumb-middle-pinky group further confirms the high similarity in brain activities elicited by index and middle finger movements. The EEGNet-based decoders obtained an average accuracy of 46.22% for the MI 4-finger task and 45.58% for the ME task, which is insufficient for real-time robotic feedback applications. Highly similar neural activation patterns between actions or motor imagery involving different fingers at the scalp level highlight the challenge in multi-finger classification (Fig. 6C, D). Interestingly, the pinky elicited the strongest EEG activation across both alpha and beta frequency bands compared to other fingers, consistent with findings from a functional near-infrared spectroscopy (fNIRS) study[83]. A plausible explanation is that the pinky requires greater attention and effort for movement or imagery, given its relatively infrequent use in daily activities[83]. Future work should explore advanced computational methods and spatial resolution enhancements to improve the control system's robustness and deepen our understanding of finger-specific neurophysiological patterns.

We investigated the contribution of distinct frequency bands to finger movement discrimination. Offline simulations revealed that broadband frequency information significantly enhanced decoding accuracy (Fig. 6E, F). Among the isolated frequency bands, the alpha band exhibited the highest performance (Fig. 6E, F), highlighting alpha oscillations as a critical frequency component for distinguishing individual finger movements. In contrast, low-frequency components showed limited discriminative power across fingers (Fig. 6E, F, Supplementary Fig. S13). This contrasts with previous studies demonstrating the utility of low-frequency MRCPs in differentiating hand and finger movements[39,64,65]. This discrepancy likely arises from the

differences in experimental design and analytical focus. MRCPs are typically observed preceding and immediately after movement onset and therefore are well-suited for characterizing discrete, non-repetitive movement tasks[84]. However, our study involved continuous decoding during repetitive finger movements or imaginations, where the transient MRCP components return to baseline shortly after the trial onsets and contribute less consistently over time. Importantly, MRCPs and ERD reflect different neural mechanisms and temporal dynamics[84]. MRCPs represent slow, time-locked potentials associated with movement planning and initiation, while alpha and beta ERD capture sustained suppression of oscillatory activity associated with ongoing motor execution or imagination. The transient nature of MRCPs limits their utility in continuous tasks, while the persistent modulation of alpha and beta rhythms throughout the movement aligns better with continuous decoding tasks such as ours. This may explain why the alpha band, in particular, showed superior performance in our classification analysis.

The inclusion of the robotic hand reflects a broader goal of this research: to enable precise, noninvasive brain-based control of physical effectors. While this study focuses on decoding individual finger movements, our long-term objective is to develop practical BCI applications for both motor-impaired and able-bodied users. Although classification naïve performance was comparable across different feedback modalities (Supplementary Fig. S19), the robotic hand serves distinct functional purposes. The virtual feedback on the screen provides binary outcome information, turning the target finger green or red depending on whether the classification was correct. In contrast, the robotic hand moves the decoded finger in real time, offering continuous and interpretable information about which finger the system believes the subject is attempting to move, even when misclassified. This enables participants to gain insight into the decoder's output. Moreover, participants reported enhanced engagement with the robotic feedback. As such, the robotic hand represents a crucial step toward achieving real-world functionality, allowing users to interact physically with their environment through brain decoding. Future work will further investigate task designs and scenarios that align more closely with everyday use cases.

Despite the wide applications of EEGNet in EEG-BCI decoding, it is a compact network designed for generalizability across various BCI paradigms. Recent developments in recurrent neural networks (RNNs), autoencoders, and transformers offer promising alternatives[45,85,86]. RNNs excel at processing time series EEG data by accounting for temporal dependencies, autoencoders learn robust latent features through data reconstruction, and transformers use attention mechanisms to enable a dynamic focus on the most discriminative parts of EEG inputs. Optimization of the decoding algorithm specifically for this finger MI design may significantly enhance the system's performance. Furthermore, integrating advanced signal processing techniques may address the current limitations in the spatial resolution of scalp EEG. Electrophysiological analysis reveals similar ERD patterns for movements and imagination of the four fingers (Fig. 6C, D), highlighting the need for better spatial resolution to distinguish between individual fingers. EEG source imaging has been demonstrated to enhance MI decoding by estimating cortical current density and exploiting information at a finer spatial scale, which especially benefits the decoding of precise MI movements[10,42,66,87,88]. Another potential approach to optimize the system performance is to combine EEG with other neuroimaging modalities with higher spatial resolution, such as fMRI. Shen et al. obtained a 63.1% average decoding accuracy across five fingers within the right hand from offline fMRI data, highlighting the spatial discrimination capability of fMRI at the level of individual fingers[89]. Future research incorporating these strategies could lead to more precise mapping of brain activity, thereby enhancing the accuracy and reliability of BCI systems and advancing the control of robotic devices.

One limitation of our study is the selection of participant groups. All participants had varying degrees of prior exposure to limb-level BCI experiments before taking part in this study, which need to be considered when interpreting the current findings. Future experiments on individuals without previous MI background may help provide a more comprehensive and generalizable insight into the paradigm. Additionally, participants were screened based on the first two offline sessions to determine if they would be considered responders instead of non-responders. It was reported previously[90] that not every individual responds well to the sensorimotor rhythm BCI paradigm, and about 15–30% participants would be considered non-responders, who do not exhibit skills to accurately control a BCI using motor imagery. We focused the present study to the responders to sensorimotor rhythm BCI, since our goal is to investigate, among BCI responders, if and how well participants can control a robotic hand at individual finger level. Under such study design, unsatisfactory offline performers were excluded from the study, so the current findings should not be interpreted as applicable to the general population. However, the increase in the subjects' engagement and motivation with continuous robotic feedback may optimize their online behavioral performance regardless of the offline results[91]. With the finger-level robotic control demonstrated in this study using a noninvasive BCI in BCI responders, future research could apply this paradigm to BCI-naïve subjects and individuals who initially perform poorly to explore its applicability to a broader target population. Another limitation of our study is that the system's performance in more intricate, real-world scenarios remains to be tested. While we demonstrated robotic control at the individual finger level using EEG-BCI, it is crucial to design BCI tasks that more closely resemble real-world applications and thoroughly evaluate the practical utility of our BCI system in these contexts.

In summary, we have demonstrated the capability of non-invasive EEG-based control for individual robotic finger movements through a naturalistic mapping between human intentions and control outputs, showcasing the potential for intuitive and sophisticated human-robot interaction. We have evaluated the efficacy of deep-learning based BCI decoder with real-time robotic feedback. Performance trends across five MI-controlled online sessions indicated that minimal training was required for the proposed paradigm. Additionally, we introduced and tested an online smoothing approach to further enhance continuous robotic control. Despite the inherent challenges of individual finger movement decoding using non-invasive techniques, the performance achieved in this study underscores the significant promise for developing more intricate and naturalistic noninvasive BCI systems. The successful demonstration of individual robotic finger control represents a substantial advancement in dexterous EEG-BCI systems and serves as a critical step forward, guiding future research in the field.

## Methods

### Brain-computer interface tasks

**Finger movement execution without feedback.** At the beginning of the study, a session of finger movement execution (ME) was conducted without feedback, meaning that participants performed the movements without receiving any visual or physical cues indicating the system's decoding results. In this session, subjects were instructed to perform repetitive single-finger flexion and extension of their right hand, including the thumb, index finger, middle finger, and pinky. This session consisted of 32 runs, each containing 5 trials for each finger in a randomized order. Each trial lasted 5 s, followed by a 2-s inter-trial interval.

**Finger motor imagery without feedback.** Following the offline finger ME session, subjects participated in a session of finger motor imagery (MI) without feedback. Subjects were instructed to imagine performing repetitive single-finger flexion and extension of their

right hand, including the thumb, index finger, middle finger, and pinky, without physically executing the movements. They were instructed to vividly imagine the sensation and kinesthetic feeling of the motion while minimizing any physical body movements, such as swallowing or gritting their teeth. To aid in maintaining their attention and engagement, subjects were encouraged to visualize a specific action involving the flexion and extension of the target finger, like pressing a piano key.

**Finger movement execution with real-time robotic feedback.** Following the initial phase of offline data collection, subjects with binary classification accuracy exceeding 70% in both offline ME and MI sessions engaged in robotic hand control through right-hand finger ME (Fig. 7A, B). This session incorporated two distinct paradigms: the binary classification paradigm, which involves distinguishing between thumb and pinky movements, and the ternary classification paradigm, which decodes a three-class condition including thumb, index finger, and pinky movements. Each online session comprised 32 runs, with each run containing 10 trials for each task in a randomized order. The first eight runs implemented the ternary classification paradigm using the base model for real-time feedback, followed by the binary classification paradigm with the base model for the next eight runs. The subsequent eight runs applied the ternary classification paradigm using the fine-tuned model for real-time feedback, and the final eight runs utilized the binary classification paradigm with the fine-tuned model (Fig. 7D).

Each trial lasted 3 s. Subjects initiated self-paced finger flexion and extension after the trial onset. The feedback period began after 1 s and continued for 2 s, during which the robotic hand moved the finger according to the latest prediction result, continuously adjusting the movements based on real-time decoding outputs. Concurrently, stimuli presented on a screen in front of the subject reflected the current online classification results by changing the color of the target finger, with green indicating correct classification and red indicating incorrect classification. A 2-s intertrial interval was appended between consecutive trials.

**Finger motor imagery with real-time robotic feedback.** Following the finger ME phase, subjects participated in MI sessions with real-time robotic feedback, which followed a design similar to the ME tasks. These sessions also incorporated two distinct paradigms: the binary classification paradigm, which differentiates between thumb and pinky imagined movements, and the ternary classification paradigm, which decodes imagined movements involving the thumb, index finger, and pinky. Each session comprised 32 runs, with each run containing 10 trials for each task in a randomized order.

The first eight runs employed the ternary classification paradigm using the base model for real-time feedback, followed by eight runs of the binary classification paradigm with the base model. The next eight runs applied the ternary classification paradigm using the fine-tuned model for real-time feedback, and the final eight runs utilized the binary classification paradigm with the fine-tuned model. Each trial lasted 3 s, with subjects initiating self-paced finger motor imagery after the trial onset. The feedback period began after 1 second and continued for 2 s. A 2-second intertrial interval was included between consecutive trials. The models for ME and MI tasks were trained separately and updated after each session to optimize the utilization of available data.

**Finger motor execution/imagery with smoothed robotic feedback.** To enhance the stability of control outputs and mitigate sensitivity to noise, an online smoothing approach was applied to both finger ME and MI tasks. This approach smoothed the outputs from the decoding models by calculating a weighted sum of current and historical probabilities, as illustrated in Eq. (1).

$$
\begin{aligned}
h_0 &= 0 \\
P_t^{'} &= \alpha * h_{t-1} + P_t \\
h_t &= P_t^{'} \\
P_t^{'} &= \frac{P_t^{'}}{||P_t^{'}||}
\end{aligned}
\tag{1}
$$

The variable $h_t$ preserved the previous probability information, initially set to $0$. The vector $P_t$ contained the raw decoding outputs for each possible class as probabilities. The smoothed output, $P_t^{'}$, was computed by combining the historical and current outputs. The parameter $\alpha$ adjusted the weighting of prior information. $P_t^{'}$ was normalized to ensure that the probabilities summed to 1. The normalized, smoothed output was subsequently used to direct the movements of the robotic hand.

To validate the efficacy of the online smoothing, one ME session and one MI session were conducted at the conclusion of this study. Each session consisted of 32 runs, with each run containing 10 trials for each task in a randomized order, similar to the online sessions described previously. In the first half of the sessions, the initial eight runs employed the ternary classification paradigm using the base model for real-time feedback, followed by eight runs utilizing the binary classification paradigm with the base model. In the latter half of the sessions, eight runs applied the ternary classification paradigm with the fine-tuned model for real-time feedback, and the final eight runs used the binary classification paradigm with the fine-tuned model. The last two sets of eight runs using the fine-tuned decoder were further divided into two groups of four runs each, one utilizing the original algorithm and the other employing the smoothing mechanism. The order of these groups was randomized, and the subjects were not informed of the paradigm changes.

**Finger motor execution/imagery with different feedback modalities.** To investigate the effects of different feedback modalities, ten subjects each participated in one ME session and one MI session. Each session consisted of 34 runs, with each run containing 10 trials for each task in a randomized order, similar to the online sessions described previously. During the first 16 runs, base models that were pre-trained using one offline session and the first two online sessions of the same subject were employed for both 2-finger and 3-finger decoding. The data collected during these initial runs was used to fine-tune the base models. In the latter half of the sessions, participants completed three runs of 2-finger and 3-finger tasks with the fine-tuned models under three conditions: visual feedback only, robotic feedback only, and combined visual and robotic feedback. The order of feedback conditions was randomized to mitigate potential bias.

## Experimental design

**Subjects.** Forty-nine right-handed able-bodied human subjects were recruited for this study. All procedures and protocols were approved by the Institutional Review Board of Carnegie Mellon University (protocol number: STUDY2017_00000548). Before participating in the experiment, subjects completed screening forms to determine their eligibility and were informed of the potential risks associated with the study. Written informed consent was obtained from all subjects prior to the commencement of the experiment. Consent for possible video recording during the experiment sessions was obtained from every subject. We did not obtain consent to publish subject-identifiable information. Ten subjects withdrew from the study due to scheduling conflicts. A uniform screening procedure was used that only subjects achieving over 70% accuracy in offline binary classification for both ME and MI tasks were included for the entire study. This screening process

was designed to address the issue that about 15–30% subjects are non-responders to sensorimotor rhythm BCI[90], and our goal is to study if and how an individual can control a robotic finger among BCI responders. Eighteen subjects were excluded from the entire study due to unsatisfactory performance in either the finger ME or MI offline sessions. The remaining twenty-one subjects (six male/fifteen female; mean age: 24.23 ± 3.72), each, completed two ME online sessions followed by two MI online sessions (Fig. 7C). Upon completion of the main study, sixteen subjects participated in three additional MI online sessions and two sessions incorporating the online smoothing (Fig. 7C).

**Experimental setup.** EEG data were acquired using a 128-channel BioSemi (BioSemi, Amsterdam, The Netherlands) EEG headcap and an ActiveTwo amplifier (BioSemi, Amsterdam, The Netherlands). A cap of appropriate size was chosen for each subject and was positioned according to the international 10–20 system for electrode placement at the beginning of each session. EEG signals were recorded at a sampling rate of 1024 Hz. During all sessions, the subjects were seated in front of a computer monitor at a distance of ~90 cm with their hands resting on an arm pillow on the desk in front of them. The robotic hand was placed in between the subjects and the screen in a position where both the visual stimuli and the robotic hand motion could be clearly observed by the subjects. Allegro Hand (Wonik Robotics, Korea), a 4-finger robotic hand with 16 degrees of freedom, was used to provide real-time visual feedback during the experiments.

**Robotic control study.** Twenty-one subjects participated in the main study to study naturalistic robotic online control for individual finger movements. All subjects had at least two prior sessions (about 2 hours of BCI training) of limb-level MI BCI experience but had no previous experience in finger-level MI. They performed one offline ME and one offline MI session for training data collection, followed by two online sessions for each task.

**Training effect of online MI.** To investigate the training effect of online MI tasks, sixteen subjects (five male/eleven female; mean age: 23.56 ± 3.22) conducted three additional online MI sessions after completing the main study. All five sessions followed the same design, with the models being updated each session by incorporating data from all previous sessions.

**Smoothed robotic control.** Sixteen subjects (five male/eleven female; mean age: 23.56 ± 3.22) participated in this phase of the study, which included one ME online session and one MI online session. In each session, subjects conducted four runs using the raw fine-tuned model outputs for robotic control and four runs using the online smoothing mechanism for both binary and ternary tasks.

### Online decoding
Online processing and classification were performed using custom Python (3.8.4) scripts made for BCPy2000 (2021.1.0), a part of the BCI2000 program[92]. The 128-channel EEG signals were re-referenced to the common average, downsampled to 100 Hz, and bandpass filtered between 4 and 40 Hz using a fourth-order Butterworth filter. The most recent 1-second segment of data was then z-scored and input to the online decoder. The decoding outputs were received by a custom C++ script for robotic finger movement control, with updates occurring every 125 ms. The finger with the highest current probability flexed its four joints by 0.1 rad within the 125 ms duration, providing visual feedback. At the end of each trial, the finger that flexed the most was considered the predicted class for that trial. All the fingers were reset to their initial position before the next trial started.

EEGNet-8,2, known for its effectiveness in EEG-BCI decoding, was employed in this study for online decoding[46]. The base model used for

online decoding during the first half of each session was pre-trained for 300 epochs on data collected in previous sessions and updated for each session. Early stopping and a learning rate scheduler were employed during the training process. The fine-tuned model used in the latter half of each session was further trained on data from the first eight runs collected on the same day. During the fine-tuning process, the parameters of the first four layers, including the temporal convolution layer and the spatial depthwise convolution layer, were fixed, while the parameters in the remaining layers were optimized based on the same-day data.

Evaluation metrics for the online decoding tasks include accuracy, precision, and recall. Accuracy is defined as the proportion of correct predictions compared to the total number of trials, where for each trial the final prediction is the one that receives the most votes from the continuous classifier outputs over multiple segments of the trial[49]. In multi-class decoding, the precision and recall are calculated for each class. Precision for a given class is the ratio of correctly classified instances of that class to the total number of instances predicted as that class by the classifier. Recall is calculated as the proportion of correct predictions to the total number of instances that actually belong to the class in the dataset.

A saliency map, which visualizes the most prominent input data that influence the model's predictions, is used to understand and interpret the inner characteristics of neural network in the finger ME and MI decoding tasks. The saliency topological map for each model is computed by summing up the gradient of the predictions with respect to the input data along the time dimension, as shown in Eq. (2).

$$X = [x_1, x_2, \ldots, x_n]$$
$$G = \frac{dP}{dX}$$
$$S_t = |\text{mean}(G)| \tag{2}$$
$$S = \sum_{t=0}^{T} S_t$$

where $x_i$ denotes a ($C \times T$) matrix representing a single input, with $C$ being the number of EEG channels and $T$ being the number of time samples in an input segment. $X$ is a ($1 \times N$) vector representing an input batch, where $N$ indicates the number of instances in the batch. $P$ is a ($N \times Z$) vector representing the predictions where $Z$ is the number of classes. $S_t$ with a dimension of ($C \times T$) is the averaged saliency map by taking the mean of $G$, the gradient of the predictions with respect to the input data, across the batch dimension. The absolute value is taken to represent the magnitude of the gradient. $S$, the spatial saliency map with a dimension of ($C \times 1$), is obtained by summing up $S_t$ along the time dimension. To show the relative importance of each channel at a group level, outliers in the saliency topological map that deviate from the mean value by greater than 2 standard deviations are removed. The Results section presents the average of the normalized saliency topological maps.

### Offline analysis
**Offline decoding.** Data collected from the offline ME and MI sessions were initially evaluated using two decoders: EEGNet-8,2 and Filter Bank Common Spatial Pattern (FBCSP) with Linear Discriminant Analysis (LDA) classifier[46,50]. Model evaluation was conducted using five-fold cross-validation. Each 5-second trial was segmented with a 1-second sliding window and a step size of 125 ms. The offline processing procedures were consistent with those used online. The offline results served as a measure of subjects' task performance and were used to exclude underperforming participants. Additionally, the offline results aided in selecting the most distinguishable 2-finger and 3-finger groups for the design of subsequent online sessions. Different frequency bands were used in bandpass filtering, including 4–40 Hz, 0.3–40 Hz, delta band (0.5–4 Hz), theta band (4–8 Hz), alpha band (8–13 Hz), and beta band (13–30 Hz), to investigate the discriminability of individual

fingers using different frequency components. Different subsets of EEG electrodes were used as inputs, including sensorimotor-region channels, non-sensorimotor channels, contralateral/ipsilateral hand knob, occipital region, frontal region, and low-density layouts (downsampled 64, 32, and 21 channels), to investigate the effects of electrode layout and cortical contributions.

**Electrophysiological analysis.** EEG data recorded during both offline and online sessions were processed for event-related desynchronization (ERD) analysis to uncover the electrophysiological patterns associated with finger MI and ME. This was done using the FieldTrip (20230118) toolbox[93] and customized MATLAB (R2023a) scripts (MathWorks Inc., MA, USA). The raw EEG data were re-referenced to the common average, downsampled to 100 Hz, and bandpass filtered between 2 and 30 Hz. Independent component analysis (ICA) was performed to remove eye movement and muscle artifacts. The continuous signal was then segmented into trials spanning from 2 s before trial onset to the end of the trial. Trials with a standard deviation above 20 μV were excluded from further analysis. To calculate ERD for each EEG channel, Morlet wavelets were used to extract the average power in the alpha (8–13 Hz) and beta (13–30 Hz) bands from 0.5 s after trial onset until the end of the trial (3 s for online sessions and 5 s for offline sessions). The average power within the same frequency bands during the 1-second period preceding trial onset was used as the baseline. Single-trial ERD was quantified by the relative change in alpha and beta band power, as calculated in Eq. (3).

$$\text{ERD}_c = \frac{(P_c - \overline{R_c})}{\overline{R_c}} \times 100\% \tag{3}$$

where $P_c$ represents the average power in the alpha and beta bands during the task period, and $\overline{R_c}$ denotes the average baseline power within each session.

Raw EEG data recorded during the offline sessions was further processed to visualize movement-related cortical potentials (MRCPs) in the low-frequency band (0.3–3 Hz). Data processing and visualization was conducted using MNE-Python (1.9.0)[94]. The EEG signals were re-referenced to the common average, downsampled to 100 Hz, and bandpass filtered between 0.3 and 3 Hz. ICA was performed to remove eye movement artifacts. The continuous data were segmented into trials, spanning from 1 s before trial onset to the end of each trial. The signal during the second before the trial onset was used for baseline correction. Group-averaged low-frequency EEG amplitude topographies were visualized for each finger. Cluster-level permutation tests were performed to identify channels and time intervals with statistically significant deviations from baseline. Additionally, group-averaged MRCPs at the selected channel (C3) were plotted for each finger movement. Time periods showing significant amplitude differences among the four fingers were determined using a one-way repeated-measures ANOVA.

**Offline simulation.** Data collected from the online ME and MI sessions were decoded using different EEGNet models, a EEGNet variant (deepEEGNet), and a FBCSP-LDA classifier to assess both machine and human learning effects. Each 3-second trial was segmented using a 1-second sliding window with a step size of 125 ms. The offline processing procedures mirrored those used during the online sessions to ensure consistency. The EEGNet models applied in the online sessions were utilized for decoding the EEG signals of interest offline, and the majority-vote accuracies were compared to the online results. Additionally, a five-fold cross-validation was conducted to evaluate the discriminability of the FBCSP-LDA classifiers on EEG signals obtained from the online sessions.

## Statistical analysis
Statistical analysis was performed using custom Python (3.10.13) scripts. For the BCI behavioral results, accuracies computed by comparing the majority vote prediction and the target finger for each trial were analyzed and compared across different conditions. Two-way repeated-measures ANOVAs were applied with main effects of session and model on the online ME and MI robotic sessions. Pairwise post hoc comparisons were performed using Wilcoxon signed-rank tests with False Discovery Rate (FDR) correction if the p value of the main effect was less than 0.05. Unless otherwise stated, Wilcoxon signed-rank tests were used to compare performance metrics, including accuracy, precision, and recall, as well as electrophysiological quantifications between two different conditions, and one-way ANOVAs were implemented to test the difference in behavioral and electrophysiological metrics under multiple conditions. Bonferroni correction was used for multiple comparisons. Two-way repeated-measures ANOVAs were employed to analyze the effects of session, model, and session-model interaction on online performance and simulation results with continued MI training. Comparison between the smoothed and original results were analyzed with linear mixed-effect models. Experimental conditions (Original versus Smoothed) were modeled as fixed effects and subjects were considered random effects.

## Ethics
Every experiment involving human participants have been carried out following a protocol approved by the Institutional Review Board of Carnegie Mellon University. Each participant gave informed written consent.

## Reporting summary
Further information on research design is available in the Nature Portfolio Reporting Summary linked to this article.

## Data availability
All data supporting the findings of this study are available within the article and its supplementary files. Any additional requests for information can be directed to, and will be fulfilled by, the corresponding authors. Source data are provided with this paper. The EEG data in all subjects used in this study are available in Figshare at: https://doi.org/10.1184/R1/29104040. Source data are provided with this paper.

## Code availability
Customized Python codes for online EEG processing and decoding as well as offline deep learning model training used in this study are available on GitHub at: https://github.com/bfinl/Finger-BCI-Decoding. The code is licensed under the MIT License.

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

## Acknowledgements
This work was supported in part by National Institutes of Health grants NS124564 (BH), NS131069 (BH), NS127849 (BH), and NSO96761 (BH). We thank Dr. Dylan Forenzo for assistance in online processing script development and discussions on statistical analysis, Simran Patibanda, Jorge Urias, and Kings Jiang for assistance in data collection, and Elena Bondi for discussion on study design.

## Author contributions
Y.D. and B.H. conceived the idea and designed the study. Y.D. and Y.Z. performed the data collection. Y.D. conducted data analysis. Y.D. wrote the original manuscript. Y.D. and B.H. revised the manuscript. Y.D. created the real-time processing scripts. C.U. built the robotic hand interface. B.H. supervised the project.

## Competing interests
The authors declare no competing interests.
