## [Transparent Peer Review file · Nature Communications]

EEG-based Brain-Computer Interface Enables Real-time Robotic Hand Control at Individual Finger Level

Corresponding Author: Dr Bin He

Version 0:

Reviewer comments:

Reviewer #1

(Remarks to the Author)

This study presents the state-of-the-art online EEG-controlled finger robotic system that functions well in two-finger and three-finger control modes. The authors hailed the superiority of a neural network model (EEGNet) in decoding either movement execution (ME) or motor imagery (MI)-based finger movement intentions from high-density EEG. With 21 healthy participants, the authors realized real-time decoding accuracies of 80.56% for two-finger control tasks and 60.61% for three-finger tasks in the MI condition. In the ME condition, the performance is similar. The results are inspiring, especially since the system only requires minimal calibration and fine-tuning. The important premise is that system performance was achieved with a group of MI-experienced participants. Whether the proposed system applies to general users or even patients remains to be explored. The manuscript is well written, with ample experiments and materials. Below, I listed my suggestions.

Major comments:

- Will the criteria of 70% accuracy in selecting responders be too strict? Since multiple sessions were tested, even naïve subjects with close to 70% accuracy are likely to gain performance improvement after training. The pre-screening step somehow limited the strength of this study, particularly in investigating the learning effects. For example, learning saturation might be different given the poor BCI performance at the beginning. In the abstract, the authors should mention that they are working on experienced subjects/BCI responders.
- According to Fig. 6, Fig. S12, and the discussion, It is a bit weird that the alpha band in the ME condition did not show significantly stronger ERD than that in the MI condition. I wonder whether this is because only experienced BCI users were included in the analysis. As the authors also recorded non-responders' EEG data, it would be nice to see whether alpha and beta band characteristics differ from the current results.
- The similar ME and MI decoding performance is confusing. As the authors noticed that ME and MI differ significantly in beta ERD strength, it is logically expected that ME has better decoding results than MI (which is more in line with existing literature). Although the authors proposed two possible explanations, I did not get why "arousal levels" and overlapping finger representations could explain this disparity. Can the authors elaborate more on this?
- What is the role of the robotic hand in this study? According to the experimental videos, the subject was given two visual feedback: one from the avatar on screen, and one from the robotic hand. There is a lack of explanation for the necessity of having a robotic system.
- Considering the authors used a high-density EEG system, would the performance drop significantly when only motor cortex electrodes were used? Discussion on the meaningful channel layout could give more guidance on future systems targeting real-world applications.
- The authors noticed an early performance plateau from multi-session experiments. One given explanation is that the decoder, which is EEGNet, is not complex enough to handle large amounts of data. Can the authors test the assumption? For instance, adding more classification layers (model parameters). I am wondering whether this plateau is due to EEG only providing limited information for finger-level decoding, or if there is a huge room for improvement.

- In the section '2.6 Alpha rhythm plays a crucial role for finger movement discrimination', the authors discussed the contribution of alpha and beta band EEG activities in finger movement discrimination. Although these bands were widely accepted as one of the most informative bands for decoding hand movements, it is also necessary to discuss other frequency bands, such as the low-frequency band (0.3 – 3Hz), particularly for ME. Studies have shown exceptional performance obtained by leveraging temporal amplitude information from low-frequency bands in decoding hand [1], finger [2], and other upper limb [3] movements.

Besides, according to Fig.6E and F, as the beta band also showed decent decoding accuracy, it is not appropriate to highlight the crucial role of alpha, especially when the broadband signals were used for EEGNet. The contribution of each frequency band to decoding performance is also related to how the features were extracted, and which decoder was used [2].

References:

[1] Ofner, P. et al. Attempted Arm and Hand Movements can be Decoded from Low-Frequency EEG from Persons with Spinal Cord Injury. *Sci Rep* 9, 7134 (2019).

[2] Sun, Q., Merino, E. C., Yang, L. & Van Hulle, M. M. Unraveling EEG correlates of unimanual finger movements: insights from non-repetitive flexion and extension tasks. *J NeuroEngineering Rehabil* 21, 228 (2024).

[3] Ofner, P., Schwarz, A., Pereira, J. & Müller-Putz, G. R. Upper limb movements can be decoded from the time-domain of low-frequency EEG. *PLoS ONE* 12, e0182578 (2017).

Minor comments:

1. Results -> 2.1 section. The first paragraph descriptions can apply to the latter sections and thus suggest that they be left out right after the Results.

2. "Performance enhancements were evident in the precision and recall metrics for individual classes, showing consistent improvements across all categories between sessions (Fig. 1C, Fig. S1A, Fig. S2)." Should the figures be Fig. 1C and D, Fig. S1A and B? Fig.S2 is unrelated to precision and recall and it is not discussed elsewhere.

3. "Our findings further support the notion that brain intentions and mental states induce changes in neuronal activation and network connectivity, which in turn generate neuronal electrical currents detectable by scalp EEG electrodes." Seems a redundant phrase.

4. "..., aligning with neurophysiological patterns of motor intention and execution (Fig. 1F, Fig. 3F, Fig. S3, Fig. S7)." Should it be Fig. 1E, Fig. 3E, Fig. S3, Fig. S7?

5. ME activates (desynchronize) more ipsilateral brain regions compared to MI. A highly relevant research would be:

Formaggio, E. et al. Modulation of event-related desynchronization in robot-assisted hand performance: brain oscillatory changes in active, passive and imagined movements. *J NeuroEngineering Rehabil* 10, 24 (2013).

Reviewer #2

(Remarks to the Author)

The paper titled "EEG-based Brain-Computer Interfaces Enable Real-time Robotic Hand Control at the Individual Finger Level" demonstrates for the first time that it is possible to control a robotic system using the execution of movement and motor imagery of individual finger movements decoded through a noninvasive EEG-based brain-computer interface. The manuscript is well-written, and the experiment and analysis are well-designed and clearly explained. The data analysis and the explanation of the results are free from procedural errors. However, I have a few minor requests to enhance the paper's readability.

1 - In the introduction section on page 4, the authors, in reference to stroke patients, assert that the "impairments significantly hinder daily living activities, underscoring the urgent need to develop a BCI-driven robotic device capable of operating at the individual finger level." However, this statement lacks a logical link between the premise and its consequences. The authors should more thoroughly explain the rationale behind their research and the existing connection between impairments in upper extremity function and the necessity for a BCI-driven robotic device controlled at the finger level. While a knowledgeable reader can easily deduce the logical connection, it should be explicitly articulated in the manuscript.

2- In the section "2.1. Finger-level robotic control via motor intention," the authors report a significant decline in decoding performance during offline decoding experiments in which EEG channels from the left sensorimotor region were masked. This result is utilized to validate the left sensorimotor region's involvement in finger MI decoding. However, there is also the possibility that the same results could be interpreted differently. Specifically, that performance declines if any brain area is masked. To rule out this interpretation, the authors should mask areas other than the left sensorimotor region, demonstrating

that they do not observe any decline in performance.

3—In the section “2.1. Finger-level robotic control via motor intention,” the authors state that “using the same input data, these improvements highlight machine learning effects.” This statement needs to be expanded to make the reader aware of the importance of these results.

4. In the section “2.1. Finger-level robotic control via motor intention,” the authors note that “the feedback period began...” They should clarify the term “feedback” even if it is already detailed in the M&M section. Additionally, in section 4.1.1, the authors immediately introduced the term feedback without explaining its meaning.

5- In the section “2.6 Alpha rhythm plays a crucial role for finger movement discrimination,” the authors comment on the results shown in Figures 6C and 6D. However, the descriptions of these figures are unclear, and it is difficult to determine when the authors are referring to the alpha band versus the beta band power. This section needs to be rewritten more clearly.

6- The same confusion is present in Figures 6C and 6D, which do not show any difference aside from the figure caption. I suggest the authors add a title to both figures so that readers can immediately understand which one represents the alpha band and which represents the beta band.

7—To improve the clarity of Figures 1C and 1D, I recommend that the authors add the figure titles (e.g., 2-class and 3-class). I also recommend moving the label box in Figure 3D to a different position that does not cover the area where the statistical analysis results should be.

Version 1:

Reviewer comments:

Reviewer #1

(Remarks to the Author)

Thank you to the authors for the detailed response to my questions. This is a solid piece of work, and I look forward to its acceptance

Reviewer #2

(Remarks to the Author)

Dear authors,

Thank you for your efforts in addressing all my comments on the manuscript. You were able to address all my comments satisfactorily.

Response to Reviewers

We express our sincere gratitude to the editors and reviewers for their constructive comments. In response, we have made substantial efforts, including additional offline simulations, online human experiments, and data analyses, and substantially revising the manuscript to address the reviewers' comments. Specifically, we conducted offline simulations to better understand the contributing factors of the proposed system, conducted two additional sessions of human experiments in a group of 10 subjects, and performed thorough statistical analyses on behaviors and associated electrophysiology data. As a result, we believe the manuscript has been significantly enhanced and improved.

Below, we provide detailed responses to each reviewer comment in **BLUE**, with the original reviewer comments in **BLACK**. Figures in the main manuscript are denoted as Fig. x, while figures in the supplemental information are denoted as Fig. Sx. Revised or added text in the manuscript is highlighted in **RED**.

Reviewer #1 (Remarks to the Author):

This study presents the state-of-the-art online EEG-controlled finger robotic system that functions well in two-finger and three-finger control modes. The authors hailed the superiority of a neural network model (EEGNet) in decoding either movement execution (ME) or motor imagery (MI)--based finger movement intentions from high-density EEG. With 21 healthy participants, the authors realized real-time decoding accuracies of 80.56% for two-finger control tasks and 60.61% for three-finger tasks in the MI condition. In the ME condition, the performance is similar. The results are inspiring, especially since the system only requires minimal calibration and fine-tuning. The important premise is that system performance was achieved with a group of MI-experienced participants. Whether the proposed system applies to general users or even patients remains to be explored. The manuscript is well written, with ample experiments and materials. Below, I listed my suggestions.

Response: We sincerely appreciate the reviewer's overall favorable assessment of our work. We have addressed specific comments from the reviewer as detailed below, including conducting additional experiments and analyses, and provided a more detailed discussion of the limitations of participant selection.

Major comments:

- Will the criteria of 70% accuracy in selecting responders be too strict? Since multiple sessions were tested, even naïve subjects with close to 70% accuracy are likely to gain performance improvement after training. The pre-screening step somehow limited the strength of this study, particularly in investigating the learning effects. For example, learning saturation might be different given the poor BCI performance at the beginning. In the abstract, the authors should mention that they are working on experienced subjects/BCI responders.

Response: Thank you for the comment. We agree with the reviewer that testing in naïve subjects or poor performers represents important research in the field of BCI that should be addressed in future investigations. We have expanded the discussion on the responders vs. non-responders in BCI research, and further clarified that in this study we aim to push the BCI performance from arm-level to finger-level robotic control in the group of responders. We have updated the **Abstract** to point out this aspect. In the **Discussion** section, we acknowledged the focus on the BCI responders as one of the limitations of the current study, and outlined future research directions,

including investigations involving naïve participants and individuals with initially low performance to better understand potential learning effects.

In **Abstract**: “In a study involving 21 able-bodied **experienced BCI users**, we achieved real-time decoding accuracies of 80.56% for two-finger MI control tasks and 60.61% for three-finger tasks. For ME-based robotic control, online performance achieved average accuracies of 81.10% for binary tasks and 60.11% for ternary tasks.”

In **Discussion**: “One limitation of our study is the selection of participant groups. All participants had varying degrees of prior exposure to limb-level BCI experiments before taking part in this study, which need to be considered when interpreting the current findings. Future experiments on individuals without previous MI background may help provide a more comprehensive and generalizable insight into the paradigm. Additionally, participants were screened based on the first two offline sessions to determine if they would be considered responders instead of non-responders. It was reported previously (90) that not every individual responds well to the sensorimotor rhythm BCI paradigm, and about 15-30% participants would be considered non-responders, who do not exhibit skills to accurately control a BCI using motor imagery. We focused the present study to the responders to sensorimotor rhythm BCI, since our goal is to investigate, among BCI responders, if and how well participants can control a robotic hand at individual finger level. Under such study design, unsatisfactory offline performers were excluded from the study, so the current findings should not be interpreted as applicable to the general population. However, the increase in the subjects’ engagement and motivation with continuous robotic feedback may optimize their online behavioral performance regardless of the offline results (91). With the finger-level robotic control demonstrated in this study using a noninvasive BCI in BCI responders, future research could apply this paradigm to BCI-naïve subjects and individuals who initially perform poorly to explore its applicability to a broader target population.”

- According to Fig. 6, Fig. S12, and the discussion, It is a bit weird that the alpha band in the ME condition did not show significantly stronger ERD than that in the MI condition. I wonder whether this is because only experienced BCI users were included in the analysis. As the authors also recorded non-responders’ EEG data, it would be nice to see whether alpha and beta band characteristics differ from the current results.

Response: Thank you for the comment. To address whether the exclusion of non-responders might influence the alpha/beta band ERD trends, we conducted additional analyses comparing responders (qualified participants) and non-responders (those excluded due to unsatisfactory performance in the beginning of study).

The averaged ME ERD topologies for the 13 subjects excluded after the offline ME sessions and the averaged ME and MI ERD topologies for the 5 subjects excluded after the offline MI sessions are visualized in Fig. S12A-D. We then compared the ERD magnitude between ME and MI conditions for both BCI responders and non-responders (Fig. S12E) and examined ERD activations between responders and non-responders across both tasks (Fig. S12F). As a result, we found that in both responders and non-responders, MI tasks consistently elicited weaker alpha and beta ERD than ME tasks (Fig. S12E), supported by medium-to-large effect sizes across groups. Statistically significant differences between ME and MI were observed only in responders, whereas non-responders showed no significance (Fig. S12E). This discrepancy may be attributed to the limited sample size of non-responders (n=5) and their inherently weaker/more variable ERD responses (Fig. S12A-D, S12F). In the original version of Fig. S12, the performances and alpha ERD from two online sessions were displayed and analyzed separately. Although a numerical difference between ME and MI alpha ERD was observed, it did not reach statistical significance.

In this revision, we performed two-way repeated-measures ANOVA on the online performances and alpha ERD, which we believe is more appropriate in our case as it better accommodates multiple factors in the data (Session and Task), and we confirmed stronger ERD in the ME conditions on the online data, consistent with our results on the offline data (updated Fig. S17). Non-responders exhibited weaker ERD magnitudes than responders in both ME and MI tasks (Fig. S12F), confirming their inability to generate sufficiently strong ERD. High inter-subject variability and small group size likely contributed to the lack of statistical significance here.

These analyses confirm that the weaker MI ERD compared to ME conditions in alpha and beta bands is consistent between responders and non-responders, reflecting inherent task-related differences in line with prior literature (Yuan et al., 2010, Lotze et al., 1999, Miller et al., 2010). We appreciate your suggestion and have incorporated these findings into the **Supplementary Material and Results**.

Yuan, H., Liu, T., Szarkowski, R., Rios, C., Ashe, J., & He, B. (2010). Negative covariation between task-related responses in alpha/beta-band activity and BOLD in human sensorimotor cortex: an EEG and fMRI study of motor imagery and movements. *NeuroImage*, 49(3), 2596–2606. <https://doi.org/10.1016/j.neuroimage.2009.10.028>

Lotze, M., Montoya, P., Erb, M., Hülsmann, E., Flor, H., Klose, U., Birbaumer, N., & Grodd, W. (1999). Activation of cortical and cerebellar motor areas during executed and imagined hand movements: an fMRI study. *Journal of cognitive neuroscience*, 11(5), 491–501. <https://doi.org/10.1162/089892999563553>

Miller, K. J., Schalk, G., Fetz, E. E., den Nijs, M., Ojemann, J. G., & Rao, R. P. (2010). Cortical activity during motor execution, motor imagery, and imagery-based online feedback. *Proceedings of the National Academy of Sciences of the United States of America*, 107(9), 4430–4435. <https://doi.org/10.1073/pnas.0913697107>

Fig. S12. Comparison of electrophysiological activities between BCI responders and non-responders. (A and B) Group-level alpha band (8 - 13 Hz) (A) and beta band (13 - 30 Hz) (B) event-related desynchronization (ERD) topographies during finger ME tasks for ME non-responders ($n = 13$). From left to right, ERD topographies corresponding to thumb, index, middle, and pinky movements are displayed. (C and D) Group-level task-specific alpha (C) and beta band (D) ERD topographies for MI non-responders ($n = 5$). The top row presents results from ME data, while the bottom row presents results from MI data. (E) Comparison of the ERD magnitude between ME and MI conditions for both BCI responders and non-responders. The center lines indicate the median value. The boxes extend from the lower quartile to the upper quartile. Diamonds indicate outliers that are more than 1.5 times the interquartile range above the third quartile or below the first quartile. Statistical analysis was conducted using a two-tailed Wilcoxon signed-rank test with Bonferroni correction (* if $p < 0.05$, n.s. if no statistical significance is found). The effect size, Cohen's d for the ERD magnitude distribution between the ME and MI conditions, is indicated under each pair of bars. (F) Comparison of the ERD magnitude between BCI responders and non-responders for both ME and MI conditions. Statistical analysis was conducted using a two-sided T-test for the means of two independent samples of scores with Bonferroni correction (n.s. if no statistical significance is found). The effect size, Cohen's d for the ERD magnitude distribution between responders and non-responders, is indicated under each pair of bars.

Fig. S17. Comparison of online performance between ME and MI. (A) Comparison between 2-finger ME online performance and 2-finger MI online performance (n = 21). The center lines indicate the median value. The boxes extend from the lower quartile to the upper quartile. The whiskers span up to 1.5 times the interquartile range. Statistical analysis using a two-way repeated-measures ANOVA with main effects of session and task (ME vs. MI). Main effect of the task: n.s. no statistical significance found. (B) Comparison between 3-finger ME online performance and 3-finger MI online performance (n = 21). Main effect of the task: n.s. no statistical significance found. (C) Comparison of alpha band ERD (8 - 13 Hz) at channel C3 for online ME task and online MI task (n = 21). Main effect of the task: # if $p < 0.05$.

- The similar ME and MI decoding performance is confusing. As the authors noticed that ME and MI differ significantly in beta ERD strength, it is logically expected that ME has better decoding results than MI (which is more in line with existing literature). Although the authors proposed two possible explanations, I did not get why “arousal levels” and overlapping finger representations could explain this disparity. Can the authors elaborate more on this?

Response: Thank you for the insightful comment. We have conducted extensive literature searches to better clarify the interplay between ERD strength, decoding performance, and the factors contributing to the comparable ME and MI results.

We visualized the temporal features extracted by EEGNet and confirmed that the model leverages subject- and task-specific narrow-band features, despite a general focus on frequency components within alpha and beta ranges (Fig. S18). Unlike the conventional ERD analysis, which relies on fixed wide-band power suppression, EEGNet is optimized to extract discriminative temporal patterns that differentiate the nuanced difference during individual finger movements and imaginations. This allows the model to detect subtle, discriminative patterns in MI that may not manifest as strong wide-band ERD but still encode task-specific information. Therefore, decoding performance is not strictly determined by ERD magnitude but by the classifier’s ability to extract task-relevant biomarkers.

Additionally, while the contralateral sensorimotor cortex plays a major role in motor decoding, our offline analysis revealed that classification performance improves when using signals from all 128 EEG channels compared to using only the contralateral motor region (Figs. S4B, S4C). Prior studies also confirmed the capability of deep learning methods in finding task-related information outside the motor cortex to aid in classification (Stieger et al., 2021), which may compensate for

weaker MI-related ERD by leveraging other task-relevant signals. Therefore, the comparable ME and MI decoding performance likely results from the deep-learning-based feature extraction from various task-relevant physiological activities, including the contralateral alpha power modulation in the motor cortex, attentional modulation, bilateral sensorimotor coordination, and other motor planning processes.

Further, the online paradigm's real-time feedback likely enhanced MI performance by modulating arousal levels. According to the Yerkes-Dodson law, optimal arousal enhances cognitive and neural engagement (Yerkes & Dodson, 1908), and neurofeedback has been shown to amplify MI-related sensorimotor rhythms beyond levels seen in ME (Miller et al., 2010) and improve classification accuracy (Hwang et al., 2009). We believe that the real-time feedback in our paradigm increased participants' arousal and attention, thereby improving the specificity of MI-related neural activity and narrowing the decoding gap between MI and ME.

Based on the findings above with supporting literature, we have included a deeper discussion in the revised manuscript as follows:

“The highly comparable performances observed between MI and ME-controlled online paradigms are likely due to enhanced engagement from real-time feedback, which helps optimize arousal levels according to the Yerkes-Dodson law, thereby enhancing cognitive and neural engagement (62, 79, 80). Neurofeedback has been shown to amplify MI-related sensorimotor rhythms beyond levels seen in ME (62) and improve classification accuracy (81). We believe that the real-time feedback in our paradigm increased participants' arousal and attention, improving the specificity of MI-related neural activity and narrowing the decoding gap between MI and ME. Furthermore, decoding performance is not strictly determined by ERD magnitude but by the classifier's ability to extract task-relevant biomarkers, including subject-specific oscillatory patterns and whole-scalp EEG distribution. EEGNet leverages subject- and task-specific narrow-band spectral features, despite a general focus on frequency components within alpha and beta ranges (Supplementary Fig. S18). Unlike the conventional ERD analysis, which relies on fixed wide-band power suppression, deep-learning-based decoders automatically identify subtle temporal patterns differentiating individual finger movements and imaginations. Such approaches can capture nuanced neural patterns that encode task-relevant information, even without pronounced ERD expression. Moreover, although contralateral sensorimotor cortex activity prominently contributes to motor decoding, incorporating whole-scalp data significantly improved classification accuracy compared to using motor-region-only signals (Supplementary Figs. S4B, S4C). Deep learning's capacity to identify distributed task-related features (44) likely compensates for weaker MI-induced ERD by integrating supplementary task-relevant physiological activities, including attentional modulation, bilateral sensorimotor coordination, and other motor planning processes. Together, these findings suggest that comparable performance between finger ME and MI arises from feedback-driven neural engagement and data-driven extraction of diverse physiological biomarkers.”

Hwang, H. J., Kwon, K., & Im, C. H. (2009). Neurofeedback-based motor imagery training for brain-computer interface (BCI). *Journal of neuroscience methods*, 179(1), 150–156. <https://doi.org/10.1016/j.jneumeth.2009.01.015>

Miller, K. J., Schalk, G., Fetz, E. E., den Nijs, M., Ojemann, J. G., & Rao, R. P. (2010). Cortical activity during motor execution, motor imagery, and imagery-based online feedback. *Proceedings of the National Academy of Sciences of the United States of America*, 107(9), 4430–4435. <https://doi.org/10.1073/pnas.0913697107>

Stieger, J. R., Engel, S. A., Suma, D., & He, B. (2021). Benefits of deep learning classification of continuous noninvasive brain-computer interface control. *Journal of neural engineering*, 18(4), 10.1088/1741-2552/ac0584. <https://doi.org/10.1088/1741-2552/ac0584>

Yerkes, R.M., & Dodson, J.D. (1908). The Relation of Strength of Stimulus to Rapidity of Habit Formation. *Journal of Comparative Neurology & Psychology*, 18, 459–482. <https://doi.org/10.1002/cne.920180503>

Fig. S18. Visualization of the temporal features derived from within-subject trained EEGNet-8,2 models for Subject 07 during the finger ME task (A), Subject 07 during the finger MI task (B), and Subject 08 during the finger ME task (C). Each row represents a learned temporal kernel from the temporal convolution layer, derived from a 1-second input window. Kernels are sorted by their dominant frequency, which is indicated to the right of each corresponding feature.

Fig. S4. Effect of EEG input from distinct cortical subregions on offline decoding performance for finger ME and MI. (A) EEG electrode layout, with channels overlying the left hand knob (dark blue), right hand knob (dark green), occipital (yellow), and frontal (dark red) regions highlighted. **(B and C)** Comparison of offline decoding accuracy between whole-scalp (All) and left hand knob (Left hand) inputs for finger ME **(B)** and MI **(C)** tasks. Error bars represent the standard error. Statistical analysis was conducted using a two-way repeated-measures ANOVA with the main effects of Finger Pairs (1-4, 1-2-4, and 1-2-3-4) and Regions (All and Left hand). Main effect of Regions: ### if $p < 0.001$. **(D and E)** Offline decoding accuracy for finger ME **(D)** and MI **(E)** tasks using EEG input from four subregions. The center lines indicate the median value. The boxes extend from the lower quartile to the upper quartile. The whiskers span up to 1.5 times the interquartile range. Statistical analysis was conducted using a two-way repeated-measures ANOVA with the main effects of Finger Pairs (1-4, 1-2-4, and 1-2-3-4) and Regions (Frontal, Left hand, Right hand, and Occipital). The ANOVA identified a significant main effect of Regions for ME ($F = 5.426$, $p = 0.002$) and MI ($F = 4.044$, $p = 0.011$) tasks. A post-hoc pairwise comparison was then performed on Channels using FDR-corrected two-tailed Wilcoxon signed-rank test results (** if $p < 0.01$, * if $p < 0.05$).

- What is the role of the robotic hand in this study? According to the experimental videos, the subject was given two visual feedback: one from the avatar on screen, and one from the robotic hand. There is a lack of explanation for the necessity of having a robotic system.

Response: Thank you for raising this important point. To clarify the role of the robotic hand, we conducted additional experiments with ten subjects to compare the effects of different feedback modalities. Each subject performed three runs of each task (ME 2-class, ME 3-class, MI 2-class, MI 3-class) under three conditions: (1) visual feedback only, (2) robotic feedback only, and (3) combined visual and robotic feedback. The order of feedback conditions was randomized to mitigate potential bias. Two-way repeated-measures ANOVA tests revealed no significant differences in performance across feedback types for both ME and MI tasks (Fig. S19).

While classification performance remained comparable, the different feedback modalities serve distinct functional purposes. The virtual feedback on the screen provides binary outcome

information—turning the target finger green or red depending on whether the classification was correct. In contrast, the robotic hand moves the decoded finger in real time, offering continuous and interpretable information about which finger the system believes the subject is attempting to move, even when misclassified. This enables participants to gain insight into the decoder’s output. Several participants also reported that the robotic feedback enhanced their engagement and helped them better understand and adapt to misclassifications.

Finally, the inclusion of the robotic hand reflects a broader goal of this research: enabling precise, noninvasive brain-based control of physical effectors. While this study focuses on decoding individual finger movements, our long-term objective is to develop practical BCI applications for both motor-impaired and able-bodied users. The robotic hand represents a crucial step toward achieving real-world functionality, allowing users to interact physically with their environment through brain decoding. Future work will further investigate task designs and scenarios that align more closely with everyday use cases.

We have incorporated further elaboration on these points in the updated **Discussion** section: “The inclusion of the robotic hand reflects a broader goal of this research: to enable precise, noninvasive brain-based control of physical effectors. While this study focuses on decoding individual finger movements, our long-term objective is to develop practical BCI applications for both motor-impaired and able-bodied users. Although classification performance was comparable across different feedback modalities (Fig. S19), the robotic hand serves distinct functional purposes. The virtual feedback on the screen provides binary outcome information, turning the target finger green or red depending on whether the classification was correct. In contrast, the robotic hand moves the decoded finger in real time, offering continuous and interpretable information about which finger the system believes the subject is attempting to move, even when misclassified. This enables participants to gain insight into the decoder’s output. Moreover, participants reported enhanced engagement with the robotic feedback. As such, the robotic hand represents a crucial step toward achieving real-world functionality, allowing users to interact physically with their environment through brain decoding. Future work will further investigate task designs and scenarios that align more closely with everyday use cases.”

Fig. S19. Online performance comparison under different feedback conditions (n = 10). Group-level online performance for 2-finger ME, 3-finger ME, 2-finger MI, and 3-finger MI tasks comparing three feedback conditions. “Both” indicates a combination of the visual feedback on the screen and the robotic feedback; “Visual” refers to visual feedback only; “Robotic” indicates robotic feedback only. The center lines indicate the median value. The boxes extend from the lower quartile to the upper quartile. The whiskers span up to 1.5 times the interquartile range. Statistical analysis was performed using a two-way repeated-measures ANOVA with the main effects of feedback and task (2-Class vs. 3-Class) on ME and MI data separately. No statistical

significance was found on the main effect of the feedback (ME: $F = 0.592$, $p = 0.556$; MI: $F = 0.105$, $p = 0.900$).

- Considering the authors used a high-density EEG system, would the performance drop significantly when only motor cortex electrodes were used? Discussion on the meaningful channel layout could give more guidance on future systems targeting real-world applications.

Response: Thank you for the excellent suggestion. Your comment prompted a thorough investigation into the trade-offs between spatial coverage, channel density, and decoding efficacy. Below, we present a detailed discussion of our findings to clarify the relationship between electrode layout, cortical contributions, and practical implications for real-world systems.

We first evaluated whether sensorimotor channels alone are sufficient for robust decoding by comparing three configurations: full 128-channel scalp coverage, only sensorimotor-region channels, and only non-sensorimotor channels. To control for the potential confounding effect of input dimensionality, the number of channels in the sensorimotor and non-sensorimotor regions was matched. Offline analyses revealed that sensorimotor channels achieved decoding performance comparable to whole-scalp input, whereas non-sensorimotor channels showed significantly reduced accuracy (updated Fig. S3). For finger MI, using only sensorimotor channels yielded significantly better performance than excluding them. This underscores the sensorimotor cortex's critical role in discriminating finger movements, consistent with its known neurophysiological function.

To further dissect cortical contributions, we isolated four subregions, each comprising 11 channels: the contralateral and ipsilateral hand knob regions, which are responsible for controlling the fine movements of fingers, occipital, and frontal regions. Among the four regions, the left and right hand knob areas showed the best results (Figs. S4D, S4E), aligning with our electrophysiological analysis results and confirming the decoder's dominant reliance on these regions. However, the accuracy using only the contralateral hand knob as input was found to be significantly lower than 128-channel whole-scalp input (Figs. S4B, S4C). This suggests that distributed scalp electrodes capture additional task-relevant information that enhances decoding robustness, such as attentional modulation, bilateral sensorimotor coordination, and downstream motor planning processes. Consistent with previous research, the results indicate deep learning's capacity to identify distributed task-related features by integrating supplementary task-relevant physiological activities (Stieger et al., 2021).

We also investigated the role of EEG channel density by simulating low-density systems (down-sampled 64, 32, and 21 channels) with broad scalp coverage. We found that performance remained stable even at 21 channels, suggesting that spatial coverage is more critical than high-density sampling (Fig. S16). This finding reflects EEG's intrinsic spatial resolution limits due to volume conduction, whereby focal activity spreads diffusely across the scalp, and also suggests that widespread activation across multiple brain regions may be involved with the motor imagery process. Consequently, sparse but spatially distributed electrodes can effectively capture the distributed neural signatures necessary for decoding, reducing system complexity and computational load without sacrificing accuracy.

In summary, while the sensorimotor cortex is central to finger movement decoding, whole-scalp coverage captures complementary neural processes that improve decoding robustness. The minimal performance drop with lower-density systems underscores that practical EEG-based BCIs can prioritize broad spatial coverage over high channel density, which provides important

guidance for designing real-world systems with reduced complexity and increased usability. The new results and discussions have been added to the **Results** and **Discussion** section.

Stieger, J. R., Engel, S. A., Suma, D., & He, B. (2021). Benefits of deep learning classification of continuous noninvasive brain-computer interface control. *Journal of neural engineering*, 18(4), 10.1088/1741-2552/ac0584. <https://doi.org/10.1088/1741-2552/ac0584>

Fig. S3. The effect of EEG input from the sensorimotor region on offline finger ME and MI decoding performance. (A) EEG electrode layout, with channels from the sensorimotor region highlighted in red. (B and C) Offline decoding results for finger ME (B) and MI (C) tasks. Decoding performance was compared using different EEG inputs. “All” refers to the condition where all 128 channels were used for decoding, “Non-SM” refers to the condition where only channels outside the sensorimotor region indicated in (A) were used for decoding, and “SM” refers to the condition where only channels inside the sensorimotor region were used for decoding. Statistical analysis was conducted using a two-way repeated-measures ANOVA with the main effects of Finger Pairs (1-4, 1-2-4, and 1-2-3-4) and Channels (All, Non-SM, and SM). The ANOVA identified a significant main effect of Channels for ME ($F = 5.785$, $p = 0.006$) and MI ($F = 5.785$, $p = 0.006$) tasks. A post-hoc pairwise comparison was then performed on Channels using FDR-corrected two-tailed Wilcoxon signed-rank test results (** if $p < 0.01$, * if $p < 0.05$). The 128-channel input significantly outperformed the non-sensorimotor region. In MI decoding, using the sensorimotor region as input yielded significantly better performance than using only the non-sensorimotor region.

Fig. S4. Effect of EEG input from distinct cortical subregions on offline decoding performance for finger ME and MI. (A) EEG electrode layout, with channels overlying the left hand knob (dark blue), right hand knob (dark green), occipital (yellow), and frontal (dark red) regions highlighted. (B and C) Comparison of offline decoding accuracy between whole-scalp (All) and left hand knob (Left hand) inputs for finger ME (B) and MI (C) tasks. Error bars represent the standard error. Statistical analysis was conducted using a two-way repeated-measures ANOVA with the main effects of Finger Pairs (1-4, 1-2-4, and 1-2-3-4) and Regions (All and Left hand). Main effect of Regions: ### if $p < 0.001$. (D and E) Offline decoding accuracy for finger ME (D) and MI (E) tasks using EEG input from four subregions. The center lines indicate the median value. The boxes extend from the lower quartile to the upper quartile. The whiskers span up to 1.5 times the interquartile range. Statistical analysis was conducted using a two-way repeated-measures ANOVA with the main effects of Finger Pairs (1-4, 1-2-4, and 1-2-3-4) and Regions (Frontal, Left hand, Right hand, and Occipital). The ANOVA identified a significant main effect of Regions for ME ($F = 5.426$, $p = 0.002$) and MI ($F = 4.044$, $p = 0.011$) tasks. A post-hoc pairwise comparison was then performed on Channels using FDR-corrected two-tailed Wilcoxon signed-rank test results (** if $p < 0.01$, * if $p < 0.05$).

Fig. S16. The effect of EEG input with different channel densities on offline finger ME and MI decoding performance. (A, B, and C) 128-channel EEG electrode layout, with 64 channels (A), 32 channels (B), and 21 channels (C) following the 10-20 international electrode layout overlaid in dark green, yellow, and dark red. (D and E) Offline decoding results for finger ME (D) and MI (E) tasks. Decoding performance was compared using EEG inputs with different channel counts, including 128 channels, in which all the recorded electrodes were used, 64 channels, 32 channels, and 21 channels, following the layouts in (A, B, and C). The center lines indicate the median value. The boxes extend from the lower quartile to the upper quartile. The whiskers span up to 1.5 times the interquartile range. A two-way repeated-measures ANOVA found no significant main effect of Channel Count for ME ($F = 1.653$, $p = 0.186$) and MI ($F = 0.296$, $p = 0.827$) tasks.

- The authors noticed an early performance plateau from multi-session experiments. One given explanation is that the decoder, which is EEGNet, is not complex enough to handle large amounts of data. Can the authors test the assumption? For instance, adding more classification layers (model parameters). I am wondering whether this plateau is due to EEG only providing limited information for finger-level decoding, or if there is a huge room for improvement.

Response: Thank you for the excellent suggestion. In response, we performed additional offline analysis using a wider and deeper EEGNet variant (deepEEGNet) by increasing the number of filters in each layer and incorporating two additional separable convolutional layers (Layer 1: 64 filters, kernel size = (1,8); Layer 2: 32 filters, kernel size = (1,16)) prior to the flattening step. Fig. S8 shows that despite a modest numerical improvement in decoding accuracy (mean (2-Class): 1.21%; mean (3-Class): 1.52%) compared to the original EEGNet, a similar learning trend was observed using deepEEGNet. Specifically, the performance plateau persisted after the initial two training sessions. This suggests that the observed trend may stem from inherent limitations in

EEG-derived features for fine-grained finger movement decoding, rather than model capacity alone.

We fully agree that further exploration is needed to disentangle these factors. Future studies could systematically evaluate state-of-the-art decoders and assess longitudinal learning patterns in naïve subjects. Such work would clarify whether performance ceilings reflect physiological constraints or methodological limitations.

We have added a discussion on this issue and the additional analysis results obtained.

In **Discussion**: “Moreover, offline decoding results using FBCSP and deepEEGNet revealed enhanced data discriminability primarily in the first session (Supplementary Fig. S8, Supplementary Fig. S9), suggesting that performance saturation likely reflects inherent limitations in EEG-derived features for precise finger movement decoding, rather than constraints imposed by model capacity.”, and “Furthermore, despite the initial investigation on expanding the architecture size, systematical evaluation of the state-of-the-art decoders may potentially enhance decoding performance over extended training periods in future applications (76).”

Fig. S8. Simulated online decoding using deepEEGNet for MI-based robotic control tasks. (A) Simulated decoding accuracy using deepEEGNet for 2-finger and 3-finger MI over five online sessions, overlaid with the simulated decoding accuracy of EEGNet for comparison (n = 16). The grey dashed line represents the chance level, and error bars indicate the standard error. (B) Pairwise comparisons of deepEEGNet decoding accuracy between consecutive sessions for 2-

finger and 3-finger MI (n = 16). The center lines indicate the median value. The boxes extend from the lower quartile to the upper quartile. The whiskers span up to 1.5 times the interquartile range. Performance improvements across sessions using deepEEGNet were statistically assessed with one-sided Wilcoxon signed-rank tests, conducted separately for each session pair and model type. Statistical significance after Bonferroni correction is denoted as *** $p < 0.001$, ** $p < 0.01$, * $p < 0.05$.

- In the section '2.6 Alpha rhythm plays a crucial role for finger movement discrimination', the authors discussed the contribution of alpha and beta band EEG activities in finger movement discrimination. Although these bands were widely accepted as one of the most informative bands for decoding hand movements, it is also necessary to discuss other frequency bands, such as the low-frequency band (0.3 – 3Hz), particularly for ME. Studies have shown exceptional performance obtained by leveraging temporal amplitude information from low-frequency bands in decoding hand [1], finger [2], and other upper limb [3] movements.

References:

[1] Ofner, P. et al. Attempted Arm and Hand Movements can be Decoded from Low-Frequency EEG from Persons with Spinal Cord Injury. *Sci Rep* 9, 7134 (2019).

[2] Sun, Q., Merino, E. C., Yang, L. & Van Hulle, M. M. Unraveling EEG correlates of unimanual finger movements: insights from non-repetitive flexion and extension tasks. *J NeuroEngineering Rehabil* 21, 228 (2024).

[3] Ofner, P., Schwarz, A., Pereira, J. & Müller-Putz, G. R. Upper limb movements can be decoded from the time-domain of low-frequency EEG. *PLoS ONE* 12, e0182578 (2017).

Response: Thank you for the comment. To address this, we evaluated decoding performance across different frequency bands, including delta (0.5–4 Hz) and theta (4–8 Hz), through offline simulations (Updated Figs. 6E and 6F). We also expanded the frequency range (0.3–40 Hz) to incorporate the delta band and repeated the comparisons with individual frequency bands (Fig. S14). Our results indicate that decoding performance dropped significantly when using only single frequency bands in offline simulations compared to that obtained using the broadband frequency range, except for the comparisons between the alpha band and 0.3–40 Hz. Furthermore, in offline simulation, using the alpha band as input significantly outperformed the delta, theta, and beta bands, and using only the beta band significantly outperformed the condition where only the delta band was used.

Since movement-related cortical potentials (MRCPs) in the low-frequency band (0.3–3 Hz) have been well used to differentiate hand and finger movements (Ofner et al., 2017; Ofner et al., 2019; Sun et al., 2024), we also visualized low-frequency EEG activity, including time-resolved amplitude topoplots and MRCPs at the selected channel (C3) (Fig. S13). All four fingers elicited significant changes in low-frequency EEG amplitudes within the first 1.5 seconds following trial onset (Figs. S13A, S13B). The amplitude modulation began in the contralateral fronto-central regions and gradually spread to the contralateral parietal area over time, in both ME and MI conditions. No significant changes were observed prior to the trial onset or after 1.5 seconds. Compared to MI, ME elicited stronger amplitude changes. MRCPs were present in both finger ME and MI, characterized by a negative deflection in the electrode potential shortly after trial onset (Figs. S13C, S13D). Although a significant difference among the fingers was observed during a brief window around the MRCP deflection, low-frequency activity across individual fingers showed considerable overlap throughout the trial.

The discrepancy between our findings and prior studies likely arises from the differences in experimental design and analytical focus. MRCPs are typically observed preceding and immediately after movement onset and therefore are well-suited for characterizing discrete, non-repetitive movement tasks (Shibasaki & Hallett, 2006). In contrast, our study involved continuous decoding during repetitive finger movements or imaginations, where the low-frequency MRCP components return to baseline shortly after the trial onsets and contribute less consistently over time. Importantly, MRCPs and ERD reflect different neural mechanisms and temporal dynamics (Shibasaki & Hallett, 2006). MRCPs represent slow, time-locked potentials associated with movement planning and initiation, while alpha and beta ERD capture sustained suppression of oscillatory activity associated with ongoing motor execution or imagination. The transient nature of MRCPs limits their utility in continuous tasks, while the persistent modulation of alpha and beta rhythms throughout the movement aligns better with continuous decoding tasks such as ours. This may explain why the alpha band, in particular, showed superior performance in our classification analysis.

We have rewritten the section and added a discussion on this issue and the additional analysis results obtained.

Ofner, P., Schwarz, A., Pereira, J., & Müller-Putz, G. R. (2017). Upper limb movements can be decoded from the time-domain of low-frequency EEG. *PloS one*, 12(8), e0182578. <https://doi.org/10.1371/journal.pone.0182578>

Ofner, P., Schwarz, A., Pereira, J., Wyss, D., Wildburger, R., & Müller-Putz, G. R. (2019). Attempted Arm and Hand Movements can be Decoded from Low-Frequency EEG from Persons with Spinal Cord Injury. *Scientific reports*, 9(1), 7134. <https://doi.org/10.1038/s41598-019-43594-9>

Shibasaki, H., & Hallett, M. (2006). What is the Bereitschaftspotential?. *Clinical neurophysiology : official journal of the International Federation of Clinical Neurophysiology*, 117(11), 2341–2356. <https://doi.org/10.1016/j.clinph.2006.04.025>

Sun, Q., Merino, E. C., Yang, L., & Van Hulle, M. M. (2024). Unraveling EEG correlates of unimanual finger movements: insights from non-repetitive flexion and extension tasks. *Journal of neuroengineering and rehabilitation*, 21(1), 228. <https://doi.org/10.1186/s12984-024-01533-4>

Fig. 6 (E and F). Comparison of offline EEGNet decoding performance for ME (E) and MI (F) tasks using EEG signals filtered with different bandpass settings (4 – 40 Hz, alpha band, beta band, delta band, theta band). The x-axis shows classification results for different frequency bands. Offline classifications were performed on thumb vs. pinky (1-4), thumb vs. index finger vs. pinky (1-2-4), and all four fingers (1-2-3-4). Two-way ANOVA was conducted across different frequency bands and finger pairs, and statistical significance was observed in all task conditions. Significance stars indicate post hoc pairwise comparison results using FDR-corrected two-tailed Wilcoxon signed-rank test (***) if $p < 0.001$, ** if $p < 0.01$, * if $p < 0.05$).

Fig. S14. Comparison of offline EEGNet decoding performance for ME (A) and MI (B) tasks using EEG signals filtered with different bandpass settings. The x-axis shows classification results for different frequency bands (0.3 – 40 Hz, alpha band, beta band, delta band, theta band). Offline classifications were performed on thumb vs. pinky (1-4), thumb vs. index finger vs. pinky (1-2-4), and all four fingers (1-2-3-4). Two-way ANOVA was conducted across different frequency bands and finger pairs, and statistical significance was observed in all task conditions. Significance stars indicate post hoc pairwise comparison results using an FDR-corrected two-tailed Wilcoxon signed-rank test (***) if $p < 0.001$, ** if $p < 0.01$, * if $p < 0.05$).

Fig. S13. Low-frequency EEG activity during finger ME and MI tasks. (A and B) Group-averaged topographical maps of low-frequency (0.3–3 Hz) EEG amplitude within the first 1.5 seconds following trial onset for finger ME (A) and MI (B). Channels showing statistically significant differences from baseline amplitude, identified via cluster-level permutation testing, are marked in black. (C and D) MRCPs at channel C3 for different finger movements during ME (C) and MI (D). The shaded area highlights a significant amplitude difference among the four fingers ($p < 0.05$, uncorrected for multiple comparisons), based on a one-way repeated-measures ANOVA.

Besides, according to Fig.6E and F, as the beta band also showed decent decoding accuracy, it is not appropriate to highlight the crucial role of alpha, especially when the broadband signals were used for EEGNet. The contribution of each frequency band to decoding performance is also related to how the features were extracted, and which decoder was used [2].

Response: Thank you for the comment. We agree that the strong beta-band decoding and the broadband nature of EEGNet’s input features complicate the interpretation of frequency-specific contributions. We significantly revised **Section 2.6** to avoid overstating alpha’s “crucial role” and instead frame it as one of multiple contributing factors.

“Frequency-specific contributions to finger discrimination

The suppression of oscillatory activities within the alpha (8 – 13 Hz) and beta (13 – 30 Hz) frequency bands in the contralateral sensorimotor region during both ME and MI tasks has been widely discussed (58, 59). Previous studies have shown that alpha and beta frequency components offer effective discriminations between limb-level MI tasks (10, 11, 18, 60). To explore the frequency-specific information encoded in EEG during finger ME and MI tasks, we extracted alpha and beta frequency components from offline EEG recordings. Task-specific ERD topologies revealed highly similar spatial patterns in both alpha and beta bands when subjects performed or imagined movements of different fingers (Figs. 6C and 6D). During ME tasks, a **bilateral power decrease in both frequency bands** was observed **over** the sensorimotor region, with the strongest suppression on the contralateral side. In contrast, MI tasks exhibited a more

localized ERD in the left sensorimotor region across both frequency bands. When comparing the modulation of EEG oscillations induced by ME and MI tasks, we observed stronger bilateral desynchronization during ME, consistent with previous findings (59, 61, 62, 63). Additionally, power modulations in the alpha and beta bands were also observed in parietal regions.

Since the analysis focused on the BCI responders after two rounds of screening based on their ME and MI performance, which might influence the group-level alpha and beta band ERD trends, we conducted analyses comparing ERD activations between qualified participants and non-responders excluded for unsatisfactory performance. Among participants excluded for poor ME performance ($n = 13$), alpha and beta ERD in the contralateral sensorimotor region remained detectable but markedly weaker than in responders (Supplementary Figs. S12A, S12B, S12F). For individuals excluded due to inadequate MI performance despite intact ME ability ($n = 5$), contralateral ERD was evident during ME tasks but absent during MI (Supplementary Figs. S12C, S12D). Across both responders and non-responders, MI tasks consistently elicited weaker alpha and beta ERD than ME tasks (Supplementary Fig. S12E), supported by medium-to-large effect sizes across groups. Non-responders also exhibited reduced ERD magnitudes compared to responders in both ME and MI tasks (Supplementary Fig. S12F), indicating their inability to generate robust sensorimotor cortical activation. These findings demonstrate that the attenuated ERD during MI relative to ME in alpha and beta bands is a consistent feature across both groups.

Beyond the alpha and beta rhythms, movement-related cortical potentials (MRCPs) in the low-frequency band (0.3–3 Hz) have been frequently used to differentiate hand and finger movements (64, 65, 39). To assess the contribution of these slow components, we visualized time-resolved amplitude topoplots and MRCP waveforms at the selected channel (C3) (Supplementary Fig. S13). All four fingers elicited significant changes in low-frequency EEG amplitudes within the first 1.5 seconds following trial onset (Supplementary Figs. S13A, S13B). The amplitude modulation began in the contralateral fronto-central regions and spread to the contralateral parietal areas over time, in both ME and MI conditions. No significant changes were observed prior to the trial onset or after 1.5 seconds. Compared to MI, ME elicited stronger amplitude changes. MRCPs were present in both finger ME and MI, characterized by a negative deflection in the electrode shortly after trial onset (Supplementary Figs. S13C, S13D). Although finger-specific differences were observed during a brief window around the MRCP deflection, low-frequency activity across individual fingers showed considerable overlap throughout the trial.

To assess the discriminability of individual finger movements across frequency bands, we filtered the EEG data into delta (0.5 – 4 Hz), theta (4 – 8 Hz), alpha, and beta bands. Each was separately input into EEGNet, and the resulting decoding accuracies were compared with those obtained from broadband (4 – 40 Hz) data. Decoding performance significantly declined when using individual frequency bands compared to the 4 – 40 Hz input, highlighting the contribution of broadband information in decoding finger movements (Figs. 6E and 6F). Among the single-band conditions, the alpha band yielded significantly higher accuracy than the delta, theta, and beta bands, while the beta band significantly outperformed the delta band. To further examine the role of low-frequency components, we expanded the broadband range to 0.3–40 Hz to include delta activity. The decoding performance of this extended broadband input remained superior to that of individual bands, and was comparable to that of the alpha band (Supplementary Fig. S14). These findings highlight the pivotal role of broadband EEG signals and specifically underscore the critical role of alpha and beta oscillations in distinguishing individual finger movements. ”

Minor comments:

1. Results -> 2.1 section. The first paragraph descriptions can apply to the latter sections and thus suggest that they be left out right after the Results.

Response: Thank you for the suggestion. The paragraph has been moved up before **Section 2.1**, as suggested.

2. “Performance enhancements were evident in the precision and recall metrics for individual classes, showing consistent improvements across all categories between sessions (Fig. 1C, Fig. S1A, Fig. S2).” Should the figures be Fig. 1C and D, Fig. S1A and B? Fig.S2 is unrelated to precision and recall and it is not discussed elsewhere.

Response: Thank you for the comment. Fig. S2 showed intuitively how the decoding capability improved across sessions and between Base and Fine-tuned models. Discussion on Fig. S2 as well as Fig. S5 (Fig. S6 in the revised manuscript) has been added to the corresponding subsections, as attached below. However, Fig. S1A compares across sessions while Fig. S1B focuses on different model types (Base vs. Fine-tuned). Therefore, Fig. S1B does not apply to this statement.

In **Section 2.1**: “Performance enhancements were evident in the precision and recall metrics for individual classes, showing consistent improvements across all categories between sessions (Fig. 1C, Supplementary Fig. S1A). Furthermore, the feature representations learned by the deep learning decoder exhibited enhanced class discriminability with increased training size across sessions, as demonstrated by visualizations of intermediate-layer activations (Supplementary Fig. S2).”

, and “Comparisons of intermediate-layer feature maps further indicated that fine-tuned models exhibited enhanced decoding capabilities relative to base models (Fig. S2).”

In **Section 2.2**: “Enhanced class separability was observed in the learned feature space across sessions, with fine-tuned models exhibiting clearer class distinction than their base model counterparts (Fig. S6).”

3. “Our findings further support the notion that brain intentions and mental states induce changes in neuronal activation and network connectivity, which in turn generate neuronal electrical currents detectable by scalp EEG electrodes.” Seems a redundant phrase.

Response: Thank you for the comment. We removed the sentence and revised the paragraph for better logical flow.

“Our findings demonstrate that deep-learning strategies can effectively resolve subtle motor-related EEG patterns obscured by the volume conduction effect (42), enabling precise real-time BCI control. The subtle differences in electrical activity caused by individual finger movements or imagined movements are challenging to detect with conventional BCI decoders due to signal smearing. The present findings demonstrate that deep learning enables efficient and automated feature extraction from EEG time series, significantly reducing the time required for signal processing compared to traditional machine-learning-based methods. This approach facilitates the online classification of minimally processed short EEG segments, allowing for real-time feedback in an online setting and ultimately improving BCI performance. Furthermore, the proposed fine-tuning paradigm enhanced BCI performance by mitigating session-to-session variability, a challenge that standard pre-trained deep learning models were unable to address.”

4. “..., aligning with neurophysiological patterns of motor intention and execution (Fig. 1F, Fig. 3F, Fig. S3, Fig. S7).” Should it be Fig. 1E, Fig. 3E, Fig. S3, Fig. S7?

Response: Thank you for catching this oversight. You are correct—the intended references should indeed point to the panels in **Fig. 1E**, **Fig. 3E**, **Fig. S3**, and **Fig. S7**, which illustrate the neurophysiological patterns of motor intention and execution. We have updated the text to reflect the correct figures in the revised manuscript.

5. ME activates (desynchronize) more ipsilateral brain regions compared to MI. A highly relevant research would be:

Formaggio, E. et al. Modulation of event-related desynchronization in robot-assisted hand performance: brain oscillatory changes in active, passive and imagined movements. *J NeuroEngineering Rehabil* 10, 24 (2013).

Response: Thank you for the excellent suggestion. We added the reference to the **Results** and **Discussion** Section as suggested.

In **Results Section 2.6**: “When comparing the modulation of EEG oscillations induced by ME and MI tasks, we observed **stronger bilateral desynchronization** during ME, consistent with previous findings (59, 61, 62, 63).”

In **Discussion**: “Various neural patterns within the ipsilateral sensorimotor region were reported and discussed across subjects and studies (58, 63, 75).”

Reviewer #2 (Remarks to the Author):

The paper titled “EEG-based Brain-Computer Interfaces Enable Real-time Robotic Hand Control at the Individual Finger Level” demonstrates for the first time that it is possible to control a robotic system using the execution of movement and motor imagery of individual finger movements decoded through a noninvasive EEG-based brain-computer interface.

The manuscript is well-written, and the experiment and analysis are well-designed and clearly explained. The data analysis and the explanation of the results are free from procedural errors. However, I have a few minor requests to enhance the paper's readability.

Response: We appreciate the reviewer's overall favorable assessment. In response to the reviewer's specific comments, we have made significant revisions to the manuscript, as outlined below.

1 - In the introduction section on page 4, the authors, in reference to stroke patients, assert that the “impairments significantly hinder daily living activities, underscoring the urgent need to develop a BCI-driven robotic device capable of operating at the individual finger level.” However, this statement lacks a logical link between the premise and its consequences. The authors should more thoroughly explain the rationale behind their research and the existing connection between impairments in upper extremity function and the necessity for a BCI-driven robotic device controlled at the finger level. While a knowledgeable reader can easily deduce the logical connection, it should be explicitly articulated in the manuscript.

Response: Thank you for the comment. We have expanded the **Introduction** to explicitly link fine motor deficits in stroke patients to the rationale for developing finger-level BCIs.

In **Introduction**, we added: “These impairments significantly hinder daily living activities that demand precise, individual finger control, which conventional rehabilitation tools and gross motor BCIs often fail to address. This gap underscores the urgent need for BCI-driven robotic devices capable of decoding and restoring finger-level dexterity, thereby enabling patients to regain functionally critical fine motor skills.”

2- In the section “2.1. Finger-level robotic control via motor intention,” the authors report a significant decline in decoding performance during offline decoding experiments in which EEG channels from the left sensorimotor region were masked. This result is utilized to validate the left sensorimotor region's involvement in finger MI decoding. However, there is also the possibility that the same results could be interpreted differently. Specifically, that performance declines if any brain area is masked. To rule out this interpretation, the authors should mask areas other than the left sensorimotor region, demonstrating that they do not observe any decline in performance.

Response: Thank you for the excellent suggestion. To address the concern that performance decline might occur when any brain area is masked, we conducted additional control analyses.

We evaluated whether sensorimotor channels alone are sufficient for robust decoding by comparing three configurations: full 128-channel scalp coverage, only sensorimotor-region channels, and only non-sensorimotor channels. To control for the potential confounding effect of input dimensionality, the number of channels in the sensorimotor and non-sensorimotor regions was matched. Offline analyses revealed that sensorimotor channels achieved decoding performance comparable to whole-scalp input, whereas non-sensorimotor channels showed significantly reduced accuracy (updated Fig. S3). For finger MI, using only sensorimotor channels

yielded significantly better performance than excluding them. This underscores the sensorimotor cortex's critical role in discriminating finger movements, consistent with its known neurophysiological function.

We further performed the offline simulation in which only EEG channels in the contralateral motor cortex hand knob, which is responsible for controlling the fine movements of fingers within the right hand, were used and compared the results with the simulated conditions where only EEG channels in the ipsilateral motor cortex hand knob, the occipital region, and the frontal lobe were used. The number of electrodes was kept the same in all the simulated conditions to prevent potential confounding effects on the results. As a result, we found that the left and right hand knob areas showed the best results among the four regions (Figs. S4D, S4E). However, the accuracy using only the contralateral hand knob as input remained significantly lower than 128-channel whole-scalp input (Figs. S4B, S4C). This finding aligns with the saliency maps of the EEGNet models, which revealed a predominant focus on the hand knob area in the left primary motor cortex with additional emphasis in the right hand knob region and parietal channels (Fig. 1F). Together, the saliency maps and subregion analyses underscore the key role of the contralateral hand knob region in decoding finger movements, while also suggesting the involvement of other coactivated regions. Consistent with previous research, the results indicate deep learning's capacity to identify distributed task-related features by integrating supplementary task-relevant physiological activities, including attentional modulation, bilateral sensorimotor coordination, and other motor planning processes (Stieger et al., 2021).

These findings underscore the sensorimotor cortex's critical role in discriminating finger movements, while suggesting that distributed scalp electrodes capture additional task-relevant information that enhances decoding robustness.

Stieger, J. R., Engel, S. A., Suma, D., & He, B. (2021). Benefits of deep learning classification of continuous noninvasive brain-computer interface control. *Journal of neural engineering*, 18(4), 10.1088/1741-2552/ac0584. <https://doi.org/10.1088/1741-2552/ac0584>

Fig. S3. The effect of EEG input from the sensorimotor region on offline finger ME and MI decoding performance. (A) EEG electrode layout, with channels from the sensorimotor region highlighted in red. (B and C) Offline decoding results for finger ME (B) and MI (C) tasks. Decoding performance was compared using different EEG inputs. “All” refers to the condition where all 128 channels were used for decoding, “Non-SM” refers to the condition where only channels outside the sensorimotor region indicated in (A) were used for decoding, and “SM” refers to the condition where only channels inside the sensorimotor region were used for decoding. Statistical analysis was conducted using a two-way repeated-measures ANOVA with the main effects of Finger Pairs (1-4, 1-2-4, and 1-2-3-4) and Channels (All, Non-SM, and SM). The ANOVA identified a significant main effect of Channels for ME ($F = 5.785$, $p = 0.006$) and MI ($F = 5.785$, $p = 0.006$) tasks. A post-hoc pairwise comparison was then performed on Channels using FDR-corrected two-tailed

Wilcoxon signed-rank test results (** if $p < 0.01$, * if $p < 0.05$). The 128-channel input significantly outperformed the non-sensorimotor region. In MI decoding, using the sensorimotor region as input yielded significantly better performance than using only the non-sensorimotor region.

Fig. S4. Effect of EEG input from distinct cortical subregions on offline decoding performance for finger ME and MI. (A) EEG electrode layout, with channels overlying the left hand knob (dark blue), right hand knob (dark green), occipital (yellow), and frontal (dark red) regions highlighted. (B and C) Comparison of offline decoding accuracy between whole-scalp (All) and left hand knob (Left hand) inputs for finger ME (B) and MI (C) tasks. Error bars represent the standard error. Statistical analysis was conducted using a two-way repeated-measures ANOVA with the main effects of Finger Pairs (1-4, 1-2-4, and 1-2-3-4) and Regions (All and Left hand). Main effect of Regions: ### if $p < 0.001$. (D and E) Offline decoding accuracy for finger ME (D) and MI (E) tasks using EEG input from four subregions. The center lines indicate the median value. The boxes extend from the lower quartile to the upper quartile. The whiskers span up to 1.5 times the interquartile range. Statistical analysis was conducted using a two-way repeated-measures ANOVA with the main effects of Finger Pairs (1-4, 1-2-4, and 1-2-3-4) and Regions (Frontal, Left hand, Right hand, and Occipital). The ANOVA identified a significant main effect of Regions for ME ($F = 5.426$, $p = 0.002$) and MI ($F = 4.044$, $p = 0.011$) tasks. A post-hoc pairwise comparison was then performed on Channels using FDR-corrected two-tailed Wilcoxon signed-rank test results (** if $p < 0.01$, * if $p < 0.05$).

3—In the section “2.1. Finger-level robotic control via motor intention,” the authors state that “using the same input data, these improvements highlight machine learning effects.” This statement needs to be expanded to make the reader aware of the importance of these results.

Response: Thank you for highlighting the need to clarify the importance of these findings. In the revised manuscript, we have expanded the text to explicitly articulate the machine learning effects. We now emphasize that the performance gains—achieved with the same input data—demonstrate the critical role of increasing training size and fine-tuning in enhancing decoding accuracy.

In **Section 2.1**, we added: “The observed improvements were achieved using the same input data across compared models, ruling out data quality as confounding factors, and therefore highlighting the machine learning effects. The enhanced performance of fine-tuned models (S1-F, S2-F) over their base counterparts (S1-B, S2-B) demonstrates the effectiveness of machine learning-driven adaptation in refining feature extraction for session-specific data. Furthermore, the superior performance of Session 2 models (S2-B, S2-F) over Session 1 models (S1-B, S1-F) suggests enhanced decoding capacities facilitated by increased training sizes.”

4. In the section “2.1. Finger-level robotic control via motor intention,” the authors note that “the feedback period began...” They should clarify the term “feedback” even if it is already detailed in the M&M section. Additionally, in section 4.1.1, the authors immediately introduced the term feedback without explaining its meaning.

Response: Thank you for the comment. We have added detailed description and explanation about the term “feedback” in the **Results** and **Materials and Methods** section to help the audience better understand the experimental paradigm.

In **Section 2**, we added “Participants received two forms of feedback reflecting the continuous decoding results, including visual feedback on a screen, where the target finger changed color to indicate the correctness of the decoding (green for correct, red for incorrect), and physical feedback from a robotic hand, which moved the detected finger in real time.” before “The feedback period began...” to better clarify what “feedback” refers to in our study.

In **Section 4.1.1**, we updated the sentence by defining the term “feedback” when introducing it for the first time: “At the beginning of the study, a session of finger movement execution (ME) was conducted without feedback, meaning that participants performed the movements without receiving any visual or physical cues indicating the system's decoding results.”

5- In the section “2.6 Alpha rhythm plays a crucial role for finger movement discrimination,” the authors comment on the results shown in Figures 6C and 6D. However, the descriptions of these figures are unclear, and it is difficult to determine when the authors are referring to the alpha band versus the beta band power. This section needs to be rewritten more clearly.

Response: Thank you for the comment. We have rewritten **Section 2.6** to provide a more precise interpretation of the results and a clearer discussion of the contributions of alpha and beta bands in finger motor tasks. Additionally, incorporating Reviewer #1's suggestions, we have expanded the discussion to include theta (4–8 Hz) and delta (0.5–4 Hz) bands, offering a more comprehensive perspective on the electrophysiological activity involved. The section has been updated as follows.

“Frequency-specific contributions to finger discrimination

The suppression of oscillatory activities within the alpha (8 – 13 Hz) and beta (13 – 30 Hz) frequency bands in the contralateral sensorimotor region during both ME and MI tasks has been widely discussed (58, 59). Previous studies have shown that alpha and beta frequency components offer effective discriminations between limb-level MI tasks (10, 11, 18, 60). To explore the frequency-specific information encoded in EEG during finger ME and MI tasks, we extracted alpha and beta frequency components from offline EEG recordings. Task-specific ERD topologies revealed highly similar spatial patterns in both alpha and beta bands when subjects performed or imagined movements of different fingers (Figs. 6C and 6D). During ME tasks, a bilateral power decrease in both frequency bands was observed over the sensorimotor region, with the strongest suppression on the contralateral side. In contrast, MI tasks exhibited a more

localized ERD in the left sensorimotor region across both frequency bands. When comparing the modulation of EEG oscillations induced by ME and MI tasks, we observed stronger bilateral desynchronization during ME, consistent with previous findings (59, 61, 62, 63). Additionally, power modulations in the alpha and beta bands were also observed in parietal regions.

Since the analysis focused on the BCI responders after two rounds of screening based on their ME and MI performance, which might influence the group-level alpha and beta band ERD trends, we conducted analyses comparing ERD activations between qualified participants and non-responders excluded for unsatisfactory performance. Among participants excluded for poor ME performance ($n = 13$), alpha and beta ERD in the contralateral sensorimotor region remained detectable but markedly weaker than in responders (Supplementary Figs. S12A, S12B, S12F). For individuals excluded due to inadequate MI performance despite intact ME ability ($n = 5$), contralateral ERD was evident during ME tasks but absent during MI (Supplementary Figs. S12C, S12D). Across both responders and non-responders, MI tasks consistently elicited weaker alpha and beta ERD than ME tasks (Supplementary Fig. S12E), supported by medium-to-large effect sizes across groups. Non-responders also exhibited reduced ERD magnitudes compared to responders in both ME and MI tasks (Supplementary Fig. S12F), indicating their inability to generate robust sensorimotor cortical activation. These findings demonstrate that the attenuated ERD during MI relative to ME in alpha and beta bands is a consistent feature across both groups.

Beyond the alpha and beta rhythms, movement-related cortical potentials (MRCPs) in the low-frequency band (0.3–3 Hz) have been frequently used to differentiate hand and finger movements (64, 65, 39). To assess the contribution of these slow components, we visualized time-resolved amplitude topoplots and MRCP waveforms at the selected channel (C3) (Supplementary Fig. S13). All four fingers elicited significant changes in low-frequency EEG amplitudes within the first 1.5 seconds following trial onset (Supplementary Figs. S13A, S13B). The amplitude modulation began in the contralateral fronto-central regions and spread to the contralateral parietal areas over time, in both ME and MI conditions. No significant changes were observed prior to the trial onset or after 1.5 seconds. Compared to MI, ME elicited stronger amplitude changes. MRCPs were present in both finger ME and MI, characterized by a negative deflection in the electrode potential shortly after trial onset (Supplementary Figs. S13C, S13D). Although finger-specific differences were observed during a brief window around the MRCP deflection, low-frequency activity across individual fingers showed considerable overlap throughout the trial.

To assess the discriminability of individual finger movements across frequency bands, we filtered the EEG data into delta (0.5 – 4 Hz), theta (4 – 8 Hz), alpha, and beta bands. Each was separately input into EEGNet, and the resulting decoding accuracies were compared with those obtained from broadband (4 – 40 Hz) data. Decoding performance significantly declined when using individual frequency bands compared to the 4 – 40 Hz input, highlighting the contribution of broadband information in decoding finger movements (Figs. 6E and 6F). Among the single-band conditions, the alpha band yielded significantly higher accuracy than the delta, theta, and beta bands, while the beta band significantly outperformed the delta band. To further examine the role of low-frequency components, we expanded the broadband range to 0.3–40 Hz to include delta activity. The decoding performance of this extended broadband input remained superior to that of individual bands, and was comparable to that of the alpha band (Supplementary Fig. S14). These findings highlight the pivotal role of broadband EEG signals and specifically underscore the critical role of alpha and beta oscillations in distinguishing individual finger movements.”

We also added explicit titles to **Fig. 6C** (Alpha ERD) and **Fig. 6D** (Beta ERD) to ensure immediate visual distinction. Please refer to the response to the next point for the updated **Fig. 6**.

6- The same confusion is present in Figures 6C and 6D, which do not show any difference aside from the figure caption. I suggest the authors add a title to both figures so that readers can immediately understand which one represents the alpha band and which represents the beta band.

Response: Thank you for the suggestion. We have added descriptive titles to **Figs. 6C, D** to explicitly denote the corresponding conditions in their respective panels, eliminating potential confusion and enhancing the readability of these figures.

Fig. 6. Offline decoding performance and electrophysiological analysis. (A and B) Group-level offline decoding performance of finger ME (A) and MI (B) using EEGNet and FBCSP ($n = 21$), where 1 represents thumb, 2 represents index finger, 3 represents middle finger, and 4 refers to pinky. The red dashed lines indicate the chance level. The center lines indicate the median value. The boxes extend from the lower quartile to the upper quartile. The whiskers span up to 1.5 times the interquartile range. (C and D) Group-level task-specific alpha band (8 - 13 Hz) (C) and beta band (13 - 30 Hz) (D) event-related desynchronization (ERD) topographies ($n = 21$). From left to right, ERD topographies corresponding to thumb, index, middle, and pinky movements are displayed. The top row presents results from ME data, while the bottom row

presents results from MI data. (**E** and **F**) Comparison of offline EEGNet decoding performance for ME (**E**) and MI (**F**) tasks using EEG signals filtered with different bandpass settings. The x-axis shows classification results for different frequency bands (4 – 40 Hz, alpha band, beta band, delta band, theta band). Offline classifications were performed on thumb vs. pinky (1-4), thumb vs. index finger vs. pinky (1-2-4), and all four fingers (1-2-3-4). Two-way ANOVA was conducted across different frequency bands and finger pairs, and statistical significance was observed in all task conditions. Significance stars indicate post hoc pairwise comparison results using an FDR-corrected two-tailed Wilcoxon signed-rank test (***) if $p < 0.001$, ** if $p < 0.01$, * if $p < 0.05$).

7—To improve the clarity of Figures 1C and 1D, I recommend that the authors add the figure titles (e.g., 2-class and 3-class). I also recommend moving the label box in Figure 3D to a different position that does not cover the area where the statistical analysis results should be.

Response: Thank you for the excellent suggestions. **Fig. 1** and **Fig. 3** have been updated by adding the titles and moving the label box.